# KVTuner: Sensitivity-Aware Layer-Wise Mixed-Precision KV Cache Quantization for Efficient and Nearly Lossless LLM Inference

Xing Li [* 1]  Zeyu Xing [* 2]  Yiming Li [1]  Linping Qu [1]  Hui-Ling Zhen [1]  Yiwu Yao [3]  Wulong Liu [1]
Sinno Jialin Pan [2]  Mingxuan Yuan [1]

## Abstract

KV cache quantization can improve Large Language Models (LLMs) inference throughput and latency in long contexts and large batch-size scenarios while preserving LLMs effectiveness. However, current methods have three unsolved issues: overlooking layer-wise sensitivity to KV cache quantization, high overhead of online fine-grained decision-making, and low flexibility to different LLMs and constraints. Therefore, we theoretically analyze the inherent correlation of layer-wise transformer attention patterns to KV cache quantization errors and study why key cache is generally more important than value cache for quantization error reduction. We further propose a simple yet effective framework KVTuner to adaptively search for the optimal hardware-friendly layer-wise KV quantization precision pairs for coarse-grained KV cache with multi-objective optimization and directly utilize the offline searched configurations during online inference. To reduce the computational cost of offline calibration, we utilize the intra-layer KV precision pair pruning and inter-layer clustering to reduce the search space. Experimental results show that we can achieve nearly lossless 3.25-bit mixed precision KV cache quantization for LLMs like Llama-3.1-8B-Instruct and 4.0-bit for sensitive models like Qwen2.5-7B-Instruct on mathematical reasoning tasks. The maximum inference throughput can be improved by 21.25% compared with KIVI-KV8 quantization over various context lengths. Our code and searched configurations are available at https://github.com/cmd2001/KVTuner.

*Equal contribution [1]Huawei Noah's Ark Lab [2]The Chinese University of Hong Kong [3]Huawei Computing Product Line. Correspondence to: Xing Li <li.xing2@huawei.com>, Zeyu Xing <zeyuxing@link.cuhk.edu.hk>, Sinno Jialin Pan <sinnopan@cuhk.edu.hk>, Mingxuan Yuan <yuan.mingxuan@huawei.com>.

*Proceedings of the 42nd International Conference on Machine Learning*, Vancouver, Canada. PMLR 267, 2025. Copyright 2025 by the author(s).

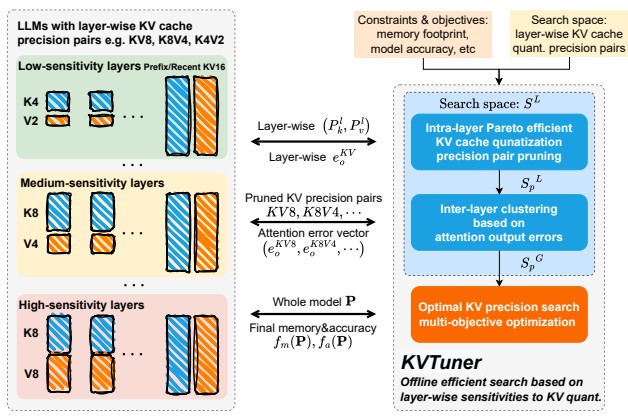

Figure 1: The layer-wise KV cache quantization tuning framework KVTuner with two-stage search space pruning for efficient MOO search using the final memory and model accuracy.

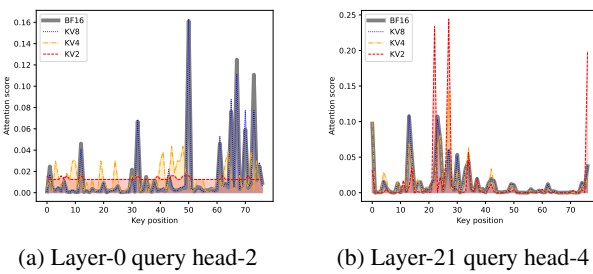

(a) Layer-0 query head-2  (b) Layer-21 query head-4

Figure 2: Token-level attention score of the 79-th query token to previous key tokens with the per-token-asym key cache quantization (Qwen2.5-7B-Instruct, GSM8K). Low-precision KV quantization (4-bit and 2-bit) causes significant distribution shifts, resulting in errors of missing or incorrect critical key identification.

## 1. Introduction

Large language models (LLMs) and multi-modality large models can comprehend and generate text, audio, image, and video like humans, showing the strong capability of assisting and interacting with humans. LLM inference efficiency such as throughput and latency is critical to enhance user experience and reduce cost. To improve the inference efficiency of LLMs, previously processed KV tokens are cached to avoid redundant recomputation. However, the memory usage of the KV cache linearly grows with the

number of batch size and sequence length, so the KV cache becomes the new bottleneck of LLM serving systems with large batching requests and long context. Valuable long context generation applications include multi-turn dialogues, long document understanding, and OpenAI o1-like level-2 reasoning. Commercial companies are releasing their supports for long context generation and KV cache-based services like prompt caching for better capability and efficiency (OpenAI, 2024; DeepSeek, 2024). Efficient KV cache management and compression can accelerate LLM inference and reduce hardware resource consumption, making it a foundational technique for advancing both enterprise-scale LLM deployment and personalized AI agents.

KV cache quantization is one of the most stable and easily deployable KV cache compression methods to reduce the memory footprint and improve throughput (Yuan et al., 2024). INT8/FP8 KV cache with dynamic asymmetric token-wise (per-token-asym) or channel-wise (per-channel-asym) quantization can achieve lossless compression in most practical applications. However, lower-bit KV cache quantization easily leads to model accuracy degradation.

Intra-layer mixed precision KV quantization methods retain important KV tokens with high precision to reduce KV cache quantization errors and quantize other cache in the same layer with uniformly low precision such as 2-bit. KIVI (Liu et al., 2024e), IntactKV (Liu et al., 2024c), and KVQuant (Hooper et al., 2024) statically keep prefix and initial KV cache blocks with high precision. They need specially designed operators for hardware like GPUs and require more careful KV cache management. Besides, the assumption that the static prefix and recent KV is important may not always hold as demonstrated in Figure 2, where low-precision quantization (4-bit and 2-bit) leads to dramatic attention distribution shift in sensitive models like Qwen2.5-7B-Instruct. Existing static and uniform KV precision methods including KIVI 4-bit cannot effectively handle these non-sparse retrieval heads. The only viable and efficient solution is to increase KV cache quantization precision of the whole model or some critical and sensitive layers.

In contrast, fine-grained methods, such as QAQ (Dong et al., 2024), MiKV (Yang et al., 2024b), and ZipCache (He et al., 2024b), dynamically identify critical KV cache and update their precision on-the-fly to improve accuracy. However, they cannot be easily integrated with flash attention (Dao et al., 2022) and vLLM (Kwon et al., 2023), because of the intra-layer fine-grained KV cache precision difference and additional deployment efforts. In addition, the online computation and control flow logic for critical token identification introduce overhead and do not fit into static graph-based inference acceleration methods.

There are still several issues to improve the inference throughput and maximum supported context length with KV cache quantization under constrained hardware resources: 1) Can we further almost losslessly compress KV cache with hardware-friendly and mixed precision quantization in a plug-and-play way? 2) Are there any other inherent model properties such as attention patterns (Tang et al., 2025; Xiao et al., 2025) that can help better trade-off memory reduction and model accuracy? 3) There are normally multiple deployed LLMs in the industrial service systems and Artificial Intelligence (AI) agents. How to adaptively tune the KV cache quantization precision considering the accuracy requirement of requests and the LLM sensitivity to KV cache quantization?

To address these issues, we thoroughly study the sensitivity of LLM transformer layers to KV cache quantization and theoretically find out that **error accumulation caused by KV cache quantization is strongly correlated with attention patterns** in Section 4.1 and 4.4. According to our observation of the sensitivity of key and value cache in the same layer in Section 4.2 and 4.3 and the layer-wise difference of transformer layers in Section 4.5, we propose to **quantize coarse-grained key and value cache in the same layer with different precision and automatically search for the optimal layer-wise KV cache quantization precision pairs based on the inherent importance of intermediate layers** in Section 5. During online serving, the offline calibrated layer-wise KV cache quantization precision pairs are directly loaded without any additional overhead to improve inference throughput and latency. Our contributions are summarized as follows:

- We study the underlying mechanism of why key cache normally is more important than value cache. The LLM accuracy degradation with low-bit key cache quantization is mainly caused by error accumulation and the layer-wise attention error distribution shift. We find out that the sensitivity of LLMs and intermediate layers to KV cache quantization is the model property and independent of input prompts.

- We propose to automatically search for the hardware-friendly layer-wise KV cache precision pairs such as K8V4 and K4V2 with multi-objective optimization (MOO) under certain memory or accuracy constraints for efficient online inference. The intra-layer pruning and inter-layer clustering are used to significantly reduce the search space and the offline tuning cost.

- We empirically demonstrate that our mixed-precision KV tuning framework KVTuner can achieve almost lossless KV cache quantization with equivalent 4-bit even 3.25-bit precision in mathematical reasoning tasks for most LLMs with 21.25% inference throughput improvement.

Table 1: The error accumulation caused by low-bit KV cache quantization (KIVI-2) potentially leads to wrong responses of mathematical reasoning tasks. The input prompt is combined with 15-shot Chain-of-Thoughts (CoTs) (Wei et al., 2023) from GSM8K training dataset for Llama2-13B-chat-hf model generation. The official KIVI CUDA kernel and code are used to generate the output.

| KV cache precision | Results with 15-th input prompt *In a dance class of 20 students, 20% enrolled in contemporary dance, 25% of the remaining enrolled in jazz dance, and the rest enrolled in hip-hop dance. What percentage of the entire students enrolled in hip-hop dance?* |
|---|---|
| FP16 | Let X be the number of students enrolled in hip-hop dance. The number of students enrolled in contemporary dance is 20% of 20 = 4. The number of students enrolled in jazz dance is 25% of the remaining = 25% of (20 - 4) = 25% of 16 = 4. Therefore, the number of students enrolled in hip-hop dance is 20 - 4 - 4 = «20-4-4=12»12 . The percentage of the entire students enrolled in hip-hop dance is 12/20 x 100% = 60% . #### 60%. |
| KIVI-4 | Let X be the number of students enrolled in hip-hop dance. The number of students enrolled in contemporary dance is 20% of 20 = 4. The number of students enrolled in jazz dance is 25% of the remaining = 25% of (20 - 4) = 25% of 16 = 4. Therefore, the number of students enrolled in hip-hop dance is 20 - 4 - 4 = «20-4-4=12»12 . The percentage of the entire students enrolled in hip-hop dance is 12/20 x 100% = 60% . #### 60%. |
| KIVI-2 | Let X be the number of students who enrolled in hip-hop dance. The number of students who enrolled in contemporary dance is 20% of 20 = 4. The number of students who enrolled in jazz dance is 25% of 16 = 4. Therefore, the total number of students who enrolled in hip-hop dance is 20 + 4 + 4 = 28 . The percentage of the entire students who enrolled in hip-hop dance is 28/20 = «28/20=14»14% . #### 14. |

## 2. Related Work

KV cache management and compression methods include paged KV cache (Kwon et al., 2023), prefilling-decoding (PD) disaggregation (Qin et al., 2024), quantization (Liu et al., 2024e;c; Hooper et al., 2024; Zhang et al., 2024c; Yang et al., 2024b; He et al., 2024b;a; Dong et al., 2024), eviction (Zhang et al., 2023; Ge et al., 2024; Liu et al., 2023; Li et al., 2024a; Adnan et al., 2024), merging (Zhang et al., 2024b; Wang et al., 2024; Wan et al., 2024; Liu et al., 2024b), low-rank decomposition (Kang et al., 2024b; Sun et al., 2024a), offloading (Sheng et al., 2023; Zhang et al., 2024a), prefetching (Lee et al., 2024b), and retrieval (Tang et al., 2024). Among them, KV cache quantization is orthogonal to most other KV cache management and compression methods, so it has been integrated with eviction, retrieval, and transferring (Tang et al., 2024; Liu et al., 2024d).

Model and activation quantization methods such as GPTQ (Frantar et al., 2022), SmoothQuant (Xiao et al., 2023), AWQ (Lin et al., 2024a), SpinQuant (Liu et al., 2024f), and QServe (Lin et al., 2024b) are also used to reduce model memory usage and inference latency with low-bit computation units. Model pruning and layer skipping reduce computational cost by directly pruning unimportant layers or heads (Ma et al., 2023; Zeng et al., 2023; Elhoushi et al., 2024).

Speculative decoding is another promising direction for lossless LLM inference acceleration by reducing the LLM inference iteration times and KV cache memory movement cost in the memory-bounded decoding stage. LLMs verify multiple tokens speculated with smaller models (Li et al., 2024b), self-partial layers (Cai et al., 2024; Liu et al., 2024a; Gloeckle et al., 2024; Stern et al., 2018), or other training-free algorithms (Zhao et al., 2024) in one forward step. In addition, Triforce (Sun et al., 2024b) is proposed to integrate KV cache compression with hierarchical speculative decoding to improve long context generation efficiency.

## 3. Background

### 3.1. Transformer and KV Cache

In LLMs, there are multiple intermediate transformer layers stacked and executed to generate final output responses. For the $l$-th transformer layer, given $i$-th D-dimensional input hidden state $\boldsymbol{x}_i^l \in \mathbb{R}^D$, the $l$-th query, key, and value feedforward neural network layers generate $\boldsymbol{q}_i^l = \boldsymbol{W}_q^l \boldsymbol{x}_i^l$, $\boldsymbol{k}_i^l = \boldsymbol{W}_k^l \boldsymbol{x}_i^l$, and $\boldsymbol{v}_i^l = \boldsymbol{W}_v^l \boldsymbol{x}_i^l$ with the corresponding weight matrices $\boldsymbol{W}_q^l$, $\boldsymbol{W}_k^l$, and $\boldsymbol{W}_v^l$, respectively. Then the self-attention scores $\boldsymbol{a}_i^l$ are computed with the current query embedding and all key embeddings until the $i$-th step. Finally, the $l$-th self-attention layer generates the output state $\boldsymbol{o}_i^l$, which is forwarded to downstream sub-layers in the $l$-th transformer layer, with the softly weighted value embeddings $\boldsymbol{V}^l$ using the attention scores $\boldsymbol{a}_i^l$:

$$\boldsymbol{a}_i^l = \text{softmax}\left(\frac{\boldsymbol{q}_i^l \boldsymbol{K}^{l\top}}{\sqrt{D}}\right), \ \boldsymbol{o}_i^l = \boldsymbol{a}_i^l \boldsymbol{V}^l, \qquad (1)$$

where $\boldsymbol{K}^l = \text{concat}(\boldsymbol{K}_{:i-1}^l, \boldsymbol{k}_i^l)$ and $\boldsymbol{V}^l = \text{concat}(\boldsymbol{V}_{:i-1}^l, \boldsymbol{v}_i^l)$ are the key and value embeddings generated in the prefilling and decoding stage in $l$-th transformer layer until $i$-th step. They will still be re-used in subsequent generation steps for self-attention computation. Therefore, we need to store them as KV cache in each layer independently to remove the additional computational cost of KV cache re-computation.

### 3.2. KV Cache Quantization

Although storing KV cache can reduce the re-computation cost, the KV cache may become the new inference memory and latency bottleneck in the large batch size and long context scenario. KV cache quantization can effectively address these problems. The round-to-nearest $B$-bit quantization and dequantization along the channel or token dimension to input $\boldsymbol{X} \in \mathbb{R}^{S \times D}$ are defined as

$$Q(\boldsymbol{X}) = \text{round}\left(\frac{\boldsymbol{X} - \boldsymbol{z}}{\boldsymbol{s}}\right), \ \hat{\boldsymbol{X}} = Q(\boldsymbol{X}) \cdot \boldsymbol{s} + \boldsymbol{z}, \quad (2)$$

where the offset $\boldsymbol{z} = \min(\boldsymbol{X})$ and the scale $\boldsymbol{s} = \frac{\max(\boldsymbol{X}) - \min(\boldsymbol{X})}{2^B - 1}$. We measure the relative KV cache and attention output errors and the absolute attention score error as $e_k^l = \text{mean}\left(\frac{|\boldsymbol{K}^l - \hat{\boldsymbol{K}}^l|}{|\boldsymbol{K}^l|}\right)$, $e_v^l = \text{mean}\left(\frac{|\boldsymbol{V}^l - \hat{\boldsymbol{V}}^l|}{|\boldsymbol{V}^l|}\right)$, $e_a^l = \text{mean}(|\boldsymbol{a}^l - \hat{\boldsymbol{a}}^l|)$, and $e_o^l = \text{mean}\left(\frac{|\boldsymbol{o}^l - \hat{\boldsymbol{o}}^l|}{|\boldsymbol{o}^l|}\right)$, where the attention score with dequantized key cache $\hat{\boldsymbol{a}}_i^l = \text{softmax}\left(\frac{\boldsymbol{q}_i^l \hat{\boldsymbol{K}}^{l\top}}{\sqrt{D}}\right)$ and the attention output with dequantized KV cache $\hat{\boldsymbol{o}}_i^l = \hat{\boldsymbol{a}}_i^l \hat{\boldsymbol{V}}^l$.

## 4. Observation

### 4.1. Error Accumulation

Due to the sequential nature of LLMs along both the model layer and token sequence dimensions, the previous layer output with KV cache quantization errors is the input of the current layer and the previous step model output token with errors is the input of the input and subsequent transformer layers. Therefore, KV cache quantization leads to two-dimensional error accumulation. The error in the $l$-th layer and $i$-th token $e_i^l$ depends on previous $1 \sim l-1$ layers and $1 \sim i-1$ steps, as defined in

$$e_i^l = f_e(\boldsymbol{e}_i^{1:l-1}, \boldsymbol{e}_{i-1}^{1:L}, \cdots, \boldsymbol{e}_1^{1:L}). \qquad (3)$$

The KV cache quantization error of a single token and layer may be ignorable. However, the error accumulation over the whole model and long context length is noticeable and may lead to token flipping and generation error, which is similar to model quantization (Lee et al., 2024a). The error accumulation caused by low-precision KV cache quantization is a general problem in domain knowledge QA, AI Generated Contents (AIGC), coding, and mathematical reasoning tasks, which may lead to critical factual errors and loss of instruction following ability.

Accumulated errors and intermediate token flipping can render the entire mathematical and logical reasoning process ineffective, resulting in unnecessary computational overhead in long-context reasoning models like OpenAI o1. As demonstrated in Table 1, KIVI-4 has exactly the same response with half-precision KV cache of an example from the GSM8K 15-shot dataset, while the first three generated sentences with low-precision KIVI-2 are highly similar to original generation except for minor differences. Additionally, there is a small token flipping from $-$ to $+$, which leads to the arithmetic operation error in the fourth sentence with KIVI-2. The wrong $20 + 4 + 4 = 28$ instead of $20 - 4 - 4 = 12$ finally leads to the arithmetic error $28/20 = $ «28/20=14»$14\%$ and the completely wrong final answer 14.

Table 2: Word-perplexity of different KV cache quantization precision pairs with the huggingface transformers KIVI-HQQ implementation on the wikitext dataset and lm-eval-harness.

| Model | KV8 | K8V4 | K8V2 | K4V8 | KV4 | K4V2 | K2V8 | K2V4 | KV2 |
|---|---|---|---|---|---|---|---|---|---|
| Llama3-8B-Instruct | 9.95 | 9.94 | 10.04 | 9.99 | 9.99 | 10.11 | 31.92 | 31.48 | 37.29 |
| Llama2-7B-chat-hf | 11.60 | 11.60 | 11.67 | 11.61 | 11.62 | 11.67 | 13.86 | 13.92 | 14.92 |
| Llama2-13B-chat-hf | 10.04 | 10.05 | 10.08 | 10.06 | 10.07 | 10.11 | 13.30 | 13.37 | 14.25 |
| Mistral-7B-Instruct-v0.3 | 8.28 | 8.27 | 8.35 | 8.31 | 8.29 | 8.44 | 12.61 | 12.71 | 15.18 |
| Qwen2.5-3B-Instruct | 10.60 | 10.59 | 11.36 | 11.11 | 11.11 | 12.28 | 147.03 | 151.30 | 251.89 |
| Qwen2.5-7B-Instruct | 9.56 | 9.39 | 9.45 | 220.83 | 235.03 | 149.15 | 1866.33 | 1831.33 | 4016.10 |
| Qwen2.5-Math-7B-Instruct | 168.92 | 169.60 | 175.34 | 588.34 | 599.02 | 725.10 | 1746.07 | 1760.31 | 1829.26 |
| Qwen2.5-14B-Instruct | 6.65 | 6.67 | 7.19 | 6.81 | 6.83 | 7.32 | 16.05 | 16.37 | 18.22 |
| Qwen2.5-32B-Instruct | 6.68 | 6.85 | 6.34 | 6.47 | 6.52 | 6.43 | 9.13 | 9.20 | 9.56 |

### 4.2. Sensitivity to Quantization Mode and Precision

KV cache quantization errors strongly correlate with the quantization mode and precision as in Table 4. In terms of relative key error $e_k$, the per-channel-asym quantization mode consistently outperforms the per-token-asym counterpart under the same precision for key cache, because key cache has strong channel-wise outliers (Liu et al., 2024e; Hooper et al., 2024), more detailed experiment results can be found in Table 9. Therefore, for specific KV cache, the quantization mode modification may lead to the shift of importance of key and value to attention output errors. As shown in Table 4, the Pareto-optimal intra-layer KV cache quantization precision pairs significantly differ between these two modes. Therefore, the KV cache precision pairs need to be adapted to quantization modes. More detailed experimental settings and results are available in Appendix B and D.1 due to space limitations.

### 4.3. Why Key Cache Is Generally More Important?

We discover the diverse model and transformer layer sensitivity to KV cache quantization mode and pairs, which is mainly caused by attention distribution shift as in Figure 2. In this section, we thus analyze the reason why key cache is normally more important than value cache from both the empirical and theoretical perspectives.

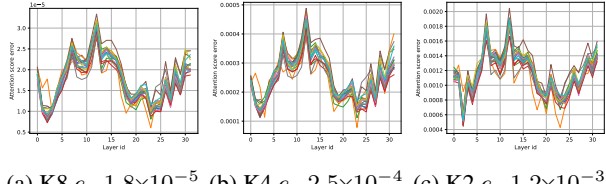

(a) K8 $e_a$ $1.8\times10^{-5}$  (b) K4 $e_a$ $2.5\times10^{-4}$  (c) K2 $e_a$ $1.2\times10^{-3}$

Figure 3: Layer-wise attention score error of per-token-asym KV cache quantization with simulated offline quantization and dequantization (without error accumulation) of the Llama-3.1-8B-Instruct model and the first 20 prompts in the zero-shot GSM8K dataset.

**Intermediate Attention Errors.** Following the settings in Table 9, we visualize the simulated layer-wise attention score errors of Llama-3.1-8B-Instruct with the per-token-

Table 3: Layer-wise relative attention output error ($e_o$) of per-token-asym KV Quant. method on Llama-3.1-8B-Instruct on the first 20 prompts from the GSM8K dataset.

| Precision | KV8 | K8V4 | K8V2 | K4V8 | KV4 | K4V2 | K2V8 | K2V4 | KV2 |
|---|---|---|---|---|---|---|---|---|---|
| Relative Attention Output Error ($e_o$) | 0.014 | 0.100 | 0.401 | 0.168 | 0.207 | 0.453 | 0.882 | 0.892 | 0.962 |

asym KV cache quantization mode in Figure 3. More results of diverse LLMs and datasets are available in Appendix F. Decreasing the key cache quantization precision from 8-bit to 4-bit and from 4-bit to 2-bit leads to $13.9\times$ and $4.6\times$ average attention score error degradation in Figure 3, respectively. It may result in attention distribution shift in the token levels of specific sensitive heads as in Figure 2 and thus degrade the final accuracy. A similar phenomenon occurs in the final output token probability when implementing KV cache eviction (Adnan et al., 2024).

As shown in Table 3, the relative attention output errors of high-precision key cache quantization with the same overall memory usage e.g. K4V2 is significantly lower than the high-precision value quantization e.g. K2V4, which empirically validates that key cache is more important than value cache during KV cache quantization of intermediate transformer layers. More detailed experiment setting and results can be found in Figure 13 and 14.

**Final Generation Errors.** We also study the final LLM generation performance with error accumulation enabled during decoding. Low-precision KV cache in all intermediate layers are quantized with the same KV precision pairs such as K8V4 and K4V2. We utilize the KIVI implementation with the HQQ backend in huggingface transformers v4.46.2 (Wolf et al., 2020), which supports popular LLMs with different scales and proposes, and measure the word-perplexity with lm-evaluation-harness (Gao et al., 2024) in Table 2.

As shown in Table 2, both KV8 and K8V4 quantization demonstrate similar perplexity levels across all models. Similarly, KV4 and K4V2 quantization demonstrate comparable patterns. These results suggest that we can achieve equivalent performance using either 6-bit (K8V4) or 3-bit (K4V2) KV cache quantization while maintaining accuracy levels similar to those of KV8 or KV4 quantization, respectively. In contrast, K4V8 and K2V4 quantizations lead to substantial increases in perplexity scores, resulting in significant degradation of generation quality. A noticeable decline in generation quality occurs when reducing the precision of the key cache rather than the value cache. The 5-bit K8V2 precision pair achieves performance equal to or better than the higher 6-bit K4V8 precision pair while achieving an additional 12.5% reduction in memory usage. These LLMs demonstrate varying levels of sensitivity to KV cache quantization. Most models experience significant perplexity increases only with int2 key cache quantization, with two no-

table exceptions: Qwen2.5-{7B, Math-7B}-Instruct. These two LLMs are sensitive even to int4 key cache quantization, indicating a lower tolerance for precision reduction. Based on these findings, we conclude that the key cache plays a more critical role than the value cache during quantization. This characteristic can be leveraged to optimize memory usage while maintaining model effectiveness.

### 4.4. Correlation of KV Quantization Errors and Attention Patterns

As shown in Figure 4, heads with high KV cache quantization errors typically exhibit non-sparse attention patterns. The sparsity patterns of the attention heads are correlated with the head-wise and layer-wise sensitivity to KV cache quantization, Highly sparse streaming heads are generally more robust to KV cache quantization than retrieval heads. The proof of Lemma 1 is available in Appendix A.

**Lemma 1.** *Only attention heads with sparse and concentrated patterns demonstrate consistent robustness to low-precision KV cache quantization.*

The optimal strategy to mitigate attention shift and enhance accuracy is to increase key quantization precision, specifically reducing $q\Delta K$ in highly sensitive layers. This approach is recommended when dynamic fine-grained token or page-level KV cache quantization for better accuracy is not feasible, as such methods remain challenging to implement on existing hardware.

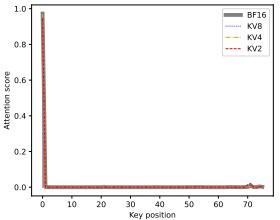 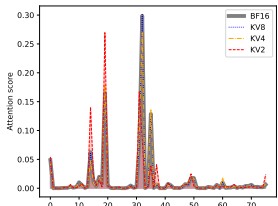

(a) Layer-2 streaming head     (b) Layer-13 retrieval head

Figure 4: Token-level attention distribution shift with the per-token-asym key cache quantization(Llama-3.1-8B-Instruct, GSM8k)

### 4.5. Layer-Wise Sensitivity to KV Cache Quantization

According to the layer-wise attention score and relative output errors of different prompts and KV cache quantization precision pairs of Llama-3.1-8B-Instruct in Figure 3 and 13, transformer layers sensitive to KV cache quantization remain consistent across different input prompts. The observed shifts in layer-wise error distribution primarily stem from variations in key cache quantization precision. Both Qwen2.5-7B-Instruct and Mistral-7B-Instruct-v0.3 exhibit similar behavioral patterns in this respect. Further analysis results can be found in Appendix F. We can thus conclude

that layer-wise sensitivity to KV cache quantization is an inherent characteristic of LLMs.

KV cache quantization errors are accumulated over both the model layer and generation sequence dimensions, and the sensitive layer will further amplify errors and lead to dramatic model performance degradation. We can perform an offline search to identify the optimal coarse-grained KV cache quantization configuration, determining the most effective precision pairs for each layer, particularly for sensitive layers, to achieve a balance between memory reduction and generation efficiency without incurring any overhead during online inference.

## 5. Method

KVTuner is an adaptive tuning framework for hardware-friendly mixed-precision KV cache quantization. It optimizes layer-wise KV precision pairs by considering their inherent sensitivity properties, aiming to achieve a better trade-off between inference efficiency and model accuracy.

Instead of making online decisions about fine-grained token or page-level KV cache quantization precision for improved model accuracy, we conduct offline search to identify the Pareto-optimal quantization precision settings for coarse-grained KV cache in each transformer layer using multi-objective optimization algorithms. Here, we refer to the entire low-bit KV cache being quantized with a specific precision pair, such as K8V4 or K4V2. This approach ensures that no additional overhead is introduced during dynamic quantization and online inference. Due to the flexibility introduced by layer-wise KV cache quantization precision tuning, KVTuner is able to accommodate more hardware and accuracy constraints of different deployed LLMs compared to uniform 8-bit or even lower precision quantization. Moreover, KVTuner accelerates LLM inference and reduces memory footprint, while still maintaining lossless or slightly lossy final model generation.

### 5.1. Problem Formulation

The offline layer-wise KV precision pair tuning problem can be formulated as a discrete combinatorial optimization task, considering hardware limitations and accuracy loss constraints. It can be solved using multi-objective optimization algorithms. We aim to minimize the quantized KV cache memory usage across all transformer layers while minimizing the final model accuracy loss, subject to the maximum $M$ memory and $\Delta A$ accuracy loss constraints:

$$\min_{\mathbf{P}} \left( f_m(\mathbf{P}), f_a(\mathbf{P}) \right) \text{ s.t. } f_m(\mathbf{P}) \leq M, f_a(\mathbf{P}) \leq \Delta A, \quad (4)$$

where the search space $\mathbf{P} \in S^L$ is the KV cache precision pairs in $L$ layers. The layer-wise search space $S$ is defined as the KV cache precision pair $(P_k^l, P_v^l)$ in the $l$-th layer.

$f_m(\mathbf{P}) = \frac{\sum(\mathbf{P})}{2L}$ captures the average equivalent quantization bits of all KV cache, $f_a(\mathbf{P}) = A_{LLM}(KV_{half}) - A_{LLM}(KV_{\mathbf{P}})$ measures the final LLM accuracy loss with the KV precision as $\mathbf{P}$ compared with LLM inference using 16-bit half precision KV cache. For instance, we can limit the average KV cache quantization precision to 2.5-bit, while optimizing the equivalent quantization precision and inference accuracy.

### 5.2. Framework

To reduce the overhead of online fine-grained KV cache mixed-precision quantization tuning, we propose offline calibration of the optimal coarse-grained KV cache quantization precision pairs for each layer or head using multi-objective optimization algorithms (Akiba et al., 2019; Zhang & Li, 2007). These pre-calibrated settings are then directly applied during online quantization. The efficiency of offline calibration is crucial for practical applications due to the large combinatorial search space of KV cache quantization pairs across multiple transformer layers. Therefore, as demonstrated in Figure 1, we propose the intra-layer and inter-layer search space pruning algorithms to accelerate the search process while preserving optimization opportunities. After the efficient preprocessing, the final LLM inference accuracy is utilized to search the Pareto optimal layer-wise KV precision pairs $\mathbf{P}$ capturing complex dependencies of the nonlinear error accumulation.

### 5.3. Automatic Layer-Wise KV Cache Quantization Precision Pair Search

As analyzed in Section 4.2 and 4.3, the model-wise and layer-wise sensitivity to KV cache quantization mode and precision is the inherent model property and is independent of the input prompts. Therefore, we can search for the optimal layer-wise KV cache quantization precision pairs offline to eliminate the additional online decision-making overhead with high generalization. If the candidate layer-wise KV precision pairs are $\{2, 4, 8\} \times \{2, 4, 8\}$, then the number of possible combinations is $9^L$, where the $L$ is the number of transformer layers. For example, the Llama-3.1-8B-Instruct model with 32 layers has about $3.4 \times 10^{30}$, which is intractable. Therefore, we design the following two-level search space pruning algorithm to reduce $\mathbf{P}$ from $S^L$ to $S_p^G$, where $S_p$ is the pruned candidate set in a group and $G$ is the number of clustered layer groups.

INTRA-LAYER KV CACHE QUANTIZATION PRECISION PAIR PRUNING

KV cache quantization errors in each layer accumulate across both the model layers and generation token dimensions. Therefore, we must control the layer-wise error by pruning KV cache quantization pairs to limit the final model

error. For all candidate KV cache quantization pairs in each layer, we prune those that are not part of the Pareto frontier, considering both the equivalent KV cache quantization precision and the relative attention output errors. For example, the precision pairs KV8, K8V4, KV4, K4V2, and KV2 are Pareto efficient for most layers in Llama-3.1-8B-Instruct in Figure 13, except for the 0-th layer, where K4V8 results in smaller errors than K8V4.

### Inter-layer Clustering

Although the above intra-layer pruning already significantly reduces the search space to $S_p^L$ such as $5^{32} \approx 2.3 \times 10^{22}$ in Llama-3.1-8B-Instruct, it is still too computationally costly for searching. Therefore, we further propose the inter-layer clustering algorithm based on relative attention output errors and the pruned candidate KV quantization pairs to $S_p^G$ such as $5^6 = 15625$. The initial step involves partitioning layers based on distinct candidate sets of pruned KV cache quantization precision pairs. These candidate sets serve as indicators of how individual layers respond differently to specific KV cache quantization precision configurations. The subsequent step involves clustering layers that share the same candidate set, using quantization sensitivity as the clustering metric. This sensitivity is quantified with the relative attention output errors produced by the pruned precision pairs.

### Calibration Dataset Design

To effectively evaluate different quantization settings, we develop an approach that amplifies KV cache quantization error accumulation and distinguishes the performance of KV precision pairs during the calibration process. This approach utilizes dequantized KV cache for self-attention computation during the prefilling stage, enabling error accumulation across model layers. Furthermore, we utilize long-context generation and challenging calibration datasets such as mathematical reasoning. In these tasks, minor errors propagating in decoding steps may result in intermediate generation token flipping and substantial mistakes in final answers as demonstrated by Table 1.

## 6. Experimental Results

The detailed experimental settings are available in Section C. The intra-layer and inter-layer KV precision pairs pruning results of various LLMs are available in Appendix D.1. The proposed pruning algorithm can significantly reduce the search space to $S_p^G$ and speedup convergence of MOO search. The final model accuracy on mathematical reasoning datasets and the throughput improvement validate the effectiveness of KVTuner.

Table 4: Intra-layer KV cache quantization precision pair pruning results of special transformer layers. The pruned Pareto efficient KV cache precision pairs in most layers are {KV8, K8V4, KV4, K4V2, KV2}, so we omit them in the table. Value is always quantized with the per-token-asym mode. $G_1$ of Mistral-7B-Instruct-v0.3 is $2\sim4$, 6, $7\sim10$, 14, 18, 27, and 29. $G_2$ of Qwen2.5-32B-Instruct is $5 \sim 10, 12, 14, 16, 18 \sim 21, 23, 26 \sim 28$, and 32.

| Model name | L | Key quant. mode | KV cache precision pairs | Layer ids |
|---|---|---|---|---|
| Llama-3.1-8B-Instruct | 32 | per-token-asym | KV8, **K4V8**, KV4, K4V2, KV2 | 0 |
| | | per-channel-asym | KV8, **K4V8**, KV4, **K2V4**, KV2 | 0 |
| | | | KV8, **K4V8**, KV4, K4V2, KV2 | 1, 2, 3, 7, 29, 31 |
| Mistral-7B-Instruct-v0.3 | 32 | per-token-asym | KV8, **K4V8**, KV4, **K2V4**, KV2 | 0 |
| | | per-channel-asym | KV8, **K4V8**, KV4, **K2V4**, KV2 | 0 |
| | | | KV8, **K4V8**, KV4, K4V2, KV2 | $G_1$ |
| Qwen2.5-3B-Instruct | 36 | per-token-asym | KV8, K8V4, *K8V2*, K4V2, KV2 | 0 |
| | | | KV8, K8V4, *K8V2*, KV4, K4V2, KV2 | 18, 27, 29 |
| | | per-channel-asym | KV8, **K4V8**, KV4, **K2V4**, KV2 | 0, 1, 2, 4, 34, 35 |
| | | | KV8, **K4V8**, KV4, K4V2, KV2 | 3, 6, 11, 13, 23 |
| Qwen2.5-7B-Instruct | 28 | per-token-asym | KV8, K8V4, *K8V2*, K4V2, KV2 | 0 |
| | | | KV8, K8V4, *K8V2*, KV4, K4V2, KV2 | 3, 13, 27 |
| | | per-channel-asym | KV8, **K4V8**, KV4, **K2V4**, KV2 | 0, 1, 2, 3 |
| | | | KV8, **K4V8**, KV4, K4V2, KV2 | 6 |
| Qwen2.5-14B-Instruct | 48 | per-token-asym | None | |
| | | per-channel-asym | KV8, **K4V8**, KV4, **K2V4**, KV2 | 0, 1, 2, 3, 4 |
| | | | KV8, **K4V8**, KV4, K4V2, KV2 | 5, 6, 8, 9, 12 |
| Qwen2.5-32B-Instruct | 64 | per-token-asym | None | |
| | | per-channel-asym | KV8, **K4V8**, KV4, **K2V4**, KV2 | 0, 1, 2, 3, 4, 11 |
| | | | KV8, **K4V8**, KV4, K4V2, KV2 | $G_2$ |
| | | | KV8, K8V4, KV4, **K2V4**, KV2 | 63 |

### 6.1. Pareto-Optimal KV Cache Precision Pair Search

**KIVI.** The mixed precision KIVI quantization mode can maintain high accuracy. As shown in Figure 5a, KVTuner with KIVI effectively maintains Llama-3.1-8B-Instruct performance while reducing the equivalent quantization precision to 3.06-bit. In addition, KVTuner also finds out four settings including lower-precision 4.91-bit in the Pareto frontier whose memory usage and accuracy are better than KV8. Most sampled settings are close to the Pareto frontier, indicating that Llama-3.1-8B-Instruct is more robust to low-precision KV quantization. These demonstrate that KVTuner increases the flexibility of KV cache quantization and can achieve lower precision and even better precision than uniform KV precision.

**Per-token-asym.** According to Figure 5b, when using the per-token-asym quantization mode on the sensitive Qwen2.5-7B-Instruct model, the Pareto frontier identified by KVTuner consistently outperforms uniform precision quantization. Especially, KVTuner can achieve KV8 accuracy with the equivalent 3.92-bit KV precision, while the uniform KV4 accuracy significantly degrades to around $0\%$. Therefore, even leveraging the simple and commonly used per-token-asym mode (Lin et al., 2024b; Sheng et al., 2023), KVTuner can reduce the memory footprint with the maintained accuracy of models with high knowledge density.

### 6.2. Mathematical and Scientific Reasoning Accuracy

Apart from the in-context few-shot GSM8K datasets, we also utilize them as the internal reasoning steps in a multi-turn way to imitate OpenAI o1 like reasoning systems in Table 5. KIVI-2 and KIVI-4 result in dramatic accuracy loss in Qwen2.5-{3B, 7B}-Instruct due to their high sensitivity

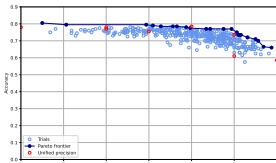

(a) Llama-3.1-8B-Instruct with KIVI

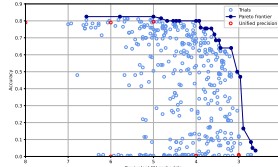

(b) Qwen2.5-7B-Instruct with per-token-asym

Figure 5: Pareto frontier on the first 200 GSM8K 4-shot prompts. Red points indicates the accuracy of 9 uniform layer-wise KV cache precision pairs including KV8, K8V4, K4V8, KV4, K4V2, and K2V4. For Qwen2.5-7B-Instruct, we can easily see that K2V8, KV4, and other lower precision pairs lose the capability of mathematical reasoning, obtaining around 0% accuracy. However, KVTuner still maintain nearly lossless overall 4-bit KV cache quantization.

Table 5: Mathematical reasoning accuracy comparison of different KV cache precision settings with the KIVI and per-token-asym quant. mode on the GSM8K few-shot CoT and CoT as multiturn dataset. We highlight the average scores with significant accuracy degradation in red and those with moderate accuracy degradation in orange. Notably, for the Qwen2.5-3B-Instruct model using KIVI quantization mode, all configurations within the 4-bit to 6-bit equivalent precision range exhibit lower accuracy on the calibration dataset compared to a configuration with an equivalent precision of 3.44-bit. As a result, we choose this 3.44-bit configuration as the highest-accuracy representative for cases where the equivalent precision is constrained to ≤6-bit.

| Quant. method | Precision | Few-shot CoT | | | Few-shot as multiturn | | | Average |
|---|---|---|---|---|---|---|---|---|
| | | 4-shot | 8-shot | 16-shot | 4-shot | 8-shot | 16-shot | |
| **Llama-3.1-8B-Instruct** | | | | | | | | |
| BF16 | BF16 | 0.7635 | 0.7741 | 0.7854 | 0.8355 | 0.8309 | 0.8332 | 0.8038 |
| Per-token-asym | KV8 | 0.7635 | 0.7710 | 0.7908 | 0.8340 | 0.8302 | 0.8279 | 0.8029 |
| | KV4 | 0.7240 | 0.7506 | 0.7354 | 0.8211 | 0.8180 | 0.8097 | 0.7765 |
| | KV2 | 0.0174 | 0.019 | 0.0250 | 0.0167 | 0.019 | 0.0197 | 0.0195 |
| | KVTuner-C5.44 | 0.7604 | 0.7726 | 0.7726 | 0.8287 | 0.8385 | 0.8309 | 0.8006 |
| | KVTuner-C3.59 | 0.7210 | 0.7316 | 0.7407 | 0.8021 | 0.8014 | 0.7991 | 0.7660 |
| KIVI | KIVI-8 | 0.7733 | 0.7748 | 0.7756 | 0.8347 | 0.8317 | 0.8294 | 0.8033 |
| | KIVI-4 | 0.7566 | 0.7718 | 0.7839 | 0.8370 | 0.8241 | 0.8332 | 0.8011 |
| | KIVI-2 | 0.6073 | 0.6080 | 0.5929 | 0.6649 | 0.6543 | 0.6687 | 0.6327 |
| | KVTuner-C4.91 | 0.7506 | 0.7665 | 0.7657 | 0.8173 | 0.8188 | 0.8378 | 0.7928 |
| | KVTuner-C3.25 | 0.7483 | 0.7566 | 0.7604 | 0.8362 | 0.8256 | 0.8279 | 0.7925 |
| **Qwen2.5-3B-Instruct** | | | | | | | | |
| BF16 | BF16 | 0.6020 | 0.6490 | 0.7020 | 0.5679 | 0.6005 | 0.6490 | 0.6284 |
| Per-token-asym | KV8 | 0.5959 | 0.6573 | 0.7081 | 0.5686 | 0.6080 | 0.6323 | 0.6284 |
| | KV4 | 0.1888 | 0.1721 | 0.2312 | 0.2229 | 0.2616 | 0.2464 | 0.2205 |
| | KV2 | 0.0099 | 0.0121 | 0.0106 | 0.0106 | 0.0091 | 0.0129 | 0.0109 |
| | KVTuner-C5.06 | 0.6058 | 0.6664 | 0.6823 | 0.5914 | 0.6133 | 0.6490 | 0.6347 |
| | KVTuner-C4.00 | 0.6156 | 0.6482 | 0.6672 | 0.5815 | 0.6118 | 0.6422 | 0.6278 |
| KIVI | KIVI-8 | 0.5974 | 0.6619 | 0.7096 | 0.5648 | 0.5989 | 0.6346 | 0.6279 |
| | KIVI-4 | 0.6156 | 0.6550 | 0.7066 | 0.5732 | 0.6073 | 0.6414 | 0.6332 |
| | KIVI-2 | 0.0546 | 0.0576 | 0.0675 | 0.047 | 0.0478 | 0.0591 | 0.0556 |
| | KVTuner-C3.44 | 0.5989 | 0.6429 | 0.7089 | 0.5701 | 0.5997 | 0.6475 | 0.6280 |
| | KVTuner-C3.17 | 0.6065 | 0.6444 | 0.6998 | 0.5512 | 0.5891 | 0.6406 | 0.6219 |
| **Qwen2.5-7B-Instruct** | | | | | | | | |
| BF16 | BF16 | 0.8059 | 0.8287 | 0.8218 | 0.7081 | 0.7501 | 0.7544 | 0.7755 |
| Per-token-asym | KV8 | 0.7998 | 0.8203 | 0.8196 | 0.7134 | 0.7384 | 0.7354 | 0.7712 |
| | KV4 | 0.0106 | 0.0121 | 0.0121 | 0.003 | 0.003 | 0.0061 | 0.0078 |
| | KV2 | 0.0068 | 0.0099 | 0.0076 | 0.0083 | 0.0106 | 0.0106 | 0.0090 |
| | KVTuner-C5.00 | 0.7885 | 0.8302 | 0.8203 | 0.6914 | 0.7445 | 0.7468 | 0.7703 |
| | KVTuner-C4.00 | 0.7847 | 0.8112 | 0.7726 | 0.6929 | 0.7331 | 0.7407 | 0.7559 |
| KIVI | KIVI-8 | 0.8021 | 0.8271 | 0.8302 | 0.7066 | 0.7354 | 0.7506 | 0.7753 |
| | KIVI-4 | 0.0735 | 0.1137 | 0.1554 | 0.0667 | 0.0705 | 0.1463 | 0.1043 |
| | KIVI-2 | 0.0379 | 0.0402 | 0.0356 | 0.0326 | 0.0258 | 0.0235 | 0.0326 |
| | KVTuner-C5.96 | 0.8218 | 0.8309 | 0.8150 | 0.6907 | 0.7248 | 0.7513 | 0.7724 |
| | KVTuner-C3.92 | 0.5959 | 0.6664 | 0.6558 | 0.5588 | 0.6156 | 0.6035 | 0.6160 |

Table 6: Scientific reasoning accuracy comparison of different KV cache precision settings with the per-token-asym KV quantization mode on the GPQA Extended dataset.

| Precision | GPQA Extended | | | Average | Precision | GPQA Extended | | | Average |
|---|---|---|---|---|---|---|---|---|---|
| | 5-shot | 10-shot | 20-shot | | | 5-shot | 10-shot | 20-shot | |
| **Llama-3.1-8B-Instruct** | | | | | **Mistral-7B-Instruct-v0.3** | | | | |
| BF16 | 0.3095 | 0.3114 | 0.2985 | 0.3065 | BF16 | 0.2930 | 0.2784 | 0.2766 | 0.2827 |
| KV8 | 0.3242 | 0.3022 | 0.3059 | 0.3108 | KV8 | 0.2985 | 0.2839 | 0.2784 | 0.2869 |
| KV4 | 0.3095 | 0.3168 | 0.3077 | 0.3113 | KV4 | 0.3040 | 0.2839 | 0.3022 | 0.2967 |
| KV2 | 0.1996 | 0.2198 | 0.2473 | 0.2222 | KV2 | 0.2857 | 0.2106 | 0.2344 | 0.2436 |
| KVTuner-C5.43 | 0.3187 | 0.3077 | 0.3187 | 0.3150 | KVTuner-C5.38 | 0.3004 | 0.2839 | 0.2912 | 0.2918 |
| KVTuner-C3.59 | 0.3223 | 0.3205 | 0.3059 | 0.3162 | KVTuner-C3.78 | 0.3260 | 0.2857 | 0.3040 | 0.3052 |
| **Qwen2.5-3B-Instruct** | | | | | **Qwen2.5-7B-Instruct** | | | | |
| BF16 | 0.3059 | 0.3095 | 0.3150 | 0.3101 | BF16 | 0.3168 | 0.3352 | 0.3297 | 0.3272 |
| KV8 | 0.3095 | 0.3059 | 0.3187 | 0.3114 | KV8 | 0.3242 | 0.3333 | 0.3407 | 0.3327 |
| KV4 | 0.2564 | 0.2711 | 0.2692 | 0.2656 | KV4 | 0.0586 | 0.0641 | 0.0751 | 0.0659 |
| KV2 | 0.0971 | 0.0806 | 0.1026 | 0.0934 | KV2 | 0.2216 | 0.1941 | 0.1996 | 0.2051 |
| KVTuner-C5.06 | 0.2985 | 0.3040 | 0.3278 | 0.3101 | KVTuner-C5.0 | 0.3315 | 0.3297 | 0.3187 | 0.3266 |
| KVTuner-C3.64 | 0.2949 | 0.3059 | 0.2985 | 0.2998 | KVTuner-C4.0 | 0.3333 | 0.3223 | 0.3205 | 0.3254 |

to low-precision KV quantization. KVTuner with KIVI can nearly losslessly quantizate KV cache to 3.92-bit, 3.17-bit, and 5.96-bit of the three models, respectively, further reducing the memory footprint compared with KIVI-4 and KIVI-8. In addition, we find out an interesting observation: KVTuner enables longer context and lower KV precision for better CoT and multi-turn mathematical reasoning accuracy than short-context and original BF16 precision KV. Most LLMs benefit from longer CoT and KVTuner enables nearly lossless lower-precision KV quantization. *We observe that KVTuner significantly reduces the performance gap between the per-token-asym and KIVI quantization modes.*

We extend our evaluation to the GPQA dataset with few-shot CoTs, as detailed in Table 6. KVTuner successfully enables lower than 4-bit, such as 3.59-bit, KV cache quantization with minimal performance degradation across various models. These results demonstrate the effectiveness of our method in maintaining high mathematical reasoning accuracy while significantly reducing memory usage.

### 6.3. Long Context Generation Accuracy

We compare KVTuner on the sensitive Qwen2.5-7B-Instruct model with the baselines KIVI-8, KIVI-4, our proposed variant KIVI-K8V4, and per-token-asym ones in the 20 Long-Bench datasets (Bai et al., 2024). The averaged scores are available in Table 7. KVTuner pushes KV cache quantization for the nearly lossless long context generation to 3.92-bit, outperforming the uniform KV precision. KVTuner with both KIVI and per-token-asym quantization methods achieve high accuracy and KV compression rates simultaneously.

### 6.4. Throughput

We measure the maximum throughput and the corresponding batch size under specific input prompt length with the implementation of the KIVI GPU kernel, which supports Llama series models. We follow the same settings and definitions of KIVI. Throughput is defined as the the number of tokens generated per second (measured end-to-end, includ-

Table 7: Accuracy comparison between offline searched layer-wise KV cache precision using KVTuner in Table 5 and 6 and uniform KV precision settings of the sensitive Qwen2.5-7B-Instruct on 20 LongBench long context generation benchmarks.

| KIVI | | | | | |
|---|---|---|---|---|---|
| BF16 | KV8 | K8V4 | KV4 | KVTuner-C5.96 | KVTuner-C3.92 |
| 0.7956 | 0.7992 | 0.8001 | 0.7723 | 0.7956 | 0.7903 |
| Per-token-asym | | | | | |
| BF16 | KV8 | K8V4 | KV4 | KVTuner-C5.0 | KVTuner-C4.0 |
| 0.7956 | 0.7971 | 0.7953 | 0.6343 | 0.8005 | 0.7960 |

ing quantization/dequantization overhead).

The layer-wise KV cache precision tuning in KVTuner are completely offline and no online overhead for precision selection is introduced. The model-level efficiency reflects the overall effects of layer-wise efficiency of all KV cache precision pairs. The memory movement cost from CPUs to GPUs and from GPU HBM to GPU cache linearly increases with the KV cache size in most case and attention is normally memory bounded. We report the total model-level throughput comparison of Llama-3.1-8B-Instruct using the searched configuration in Table 5 as below. Compared with KIVI-KV8, KVTuner-C3.25 can improve decoding throughput by 16.79%~21.25%. More efficient KV dequantization and attention kernels proposed in Qserve (Lin et al., 2024b) and TurboAttention (Kang et al., 2024a) may further enhance the throughout benefit of KVTuner than INT8 KV baselines.

Table 8: Throughput comparison between offline searched layer-wise KV cache precision using KVTuner in Table 5 and uniform KV precision settings with KIVI of Llama-3.1-8B-Instruct.

| BS | inputLen | KV8(baseline) | K8V4 | KV4 | K4V2 | KVTuner-C4.91 | KVTuner-C3.25 |
|---|---|---|---|---|---|---|---|
| 64 | 128 | 3836 | 4193 | 4567 | 4697 | 4240 +10.53% | 4652 +21.25% |
| 16 | 512 | 1102 | 1205 | 1275 | 1304 | 1239 +12.41% | 1296 +17.55% |
| 8 | 1024 | 549 | 597 | 632 | 645 | 600 +9.22% | 641 +16.79% |

### 6.5. Detailed Analysis

By analyzing the detailed configurations in the Pareto frontier identified for Llama-3.1-8B-Instruct, we observe that:

- In most cases, all layer groups adopt a quantization configuration where the precision of the key is higher than the precision of the value. This supports our earlier observation from uniform quantization that the key plays a more critical role in quantization.

- In other cases, in certain specialized layer groups, the value is set at a higher precision than the key for certain specialized layer groups. This aligns with the patterns identified in Table 4, which highlight specific layer groups may require higher precision for values.

- KVTuner tends to allocate higher precision to groups

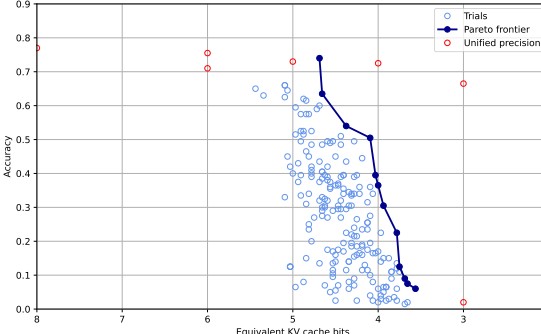

Figure 6: Pareto frontier of Llama-3.1-8B-Instruct with the per-token-asym KV quantization mode and without the proposed two-stage search space pruning on the first 200 GSM8k 4-shot prompts.

with larger quantization errors. Reducing the quantization precision of the key for a crucial group of layers can significantly degrade the performance. For instance, in Llama-3.1-8B-Instruct, the layer group [8 ∼ 11, 14 ∼ 17, 20, 30] is particularly sensitive to the reduction of the key precision, and if the precision of the key is reduced from 4-bit to 2-bit, the performance would drop from 0.67 to 0.495.

### 6.6. Ablation Studies

According to Figure 6, when using the per-token-asym quantization mode on the Llama-3.1-8B-Instruct model, the search results deteriorate significantly if the proposed intra-layer and inter-layer search space pruning algorithms are not applied. In comparison with the counterpart with search space pruning as pre-processing in Figure 9a, this highlights search space pruning is helpful for MOO search convergence and maintaining quantization performance.

## 7. Conclusion

KVTuner enables efficient and adaptive layer-wise mixed-precision KV cache quantization via sensitivity-aware optimization techniques. It systematically reduces KV cache quantization errors by prioritizing key cache precision while balancing memory efficiency and inference accuracy. Experimental results demonstrate that KVTuner achieves nearly lossless compression at 3.25-bit for Llama-3.1-8B-Instruct and 4-bit for sensitive Qwen2.5-7B-Instruct. KVTuner also demonstrates that employing longer CoTs with lower and mixed precision KV cache quantization yields superior performance compared to shorter CoTs utilizing higher precision KV cache. This improvement is evident in both memory efficiency and accuracy, particularly in the context of mathematical reasoning tasks. KVTuner also greatly narrows the performance difference between the simple per-token-asym and accurate KIVI quantization modes, even when using overall similar low-precision settings.

## Impact Statement

This paper thoroughly studies the layer-wise sensitivity of transformers to KV cache quantization methods, which is the inherent property of LLMs. Low-precision KV cache quantization may lead to significantly token-level attention distribution shift in heads with non-sparse and non-concentrated attention patterns. The attention head related property may also be applied to LLM weight and activation quantization and other KV cache compression fields. The proposed automatic KV cache precision pairs tuning algorithm makes inference acceleration of LLMs with low-precision KV cache possible, which can help reduce the deployment cost and carbon footprint. Low-precision KV cache quantization with ignorable LLM accuracy loss is an important direction to reduce the KV cache memory usage and cost in online inference, KV cache offloading (Sheng et al., 2023; Zhang et al., 2024a), storage (Jin et al., 2024), transferring (Liu et al., 2024d), and more LLM inference related applications.

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

## A. Proof of Lemma 1

Lemma 1 claims that only attention heads with sparse and concentrated patterns demonstrate consistent robustness to low-precision KV cache quantization. Its proof is below.

*Proof.* Given the query token $\boldsymbol{q} \in \mathbb{R}^{1 \times D}$ and key cache $\boldsymbol{K} \in \mathbb{R}^{D \times S}$, the attention score without errors is $a_i = \frac{\exp(\boldsymbol{q}\boldsymbol{K}_i)}{\sum_{j=1}^{S} \exp(\boldsymbol{q}\boldsymbol{K}_j)}$. The key asymmetric uniform quantization error $\Delta\boldsymbol{K} \in \mathbb{R}^{S \times D} \sim \mathcal{N}(0, \sigma^2)$ follows normal distribution, where $\sigma = \frac{max(\boldsymbol{K}) - min(\boldsymbol{K})}{2^B - 1}$. Therefore, low precision quantization leads to exponential larger quantization errors. Then, the $i$-th attention score with key errors is

$$\hat{a}_i = \frac{\exp(\boldsymbol{q}(\boldsymbol{K}_i + \Delta\boldsymbol{K}_i))}{\sum_{j=1}^{S} \exp(\boldsymbol{q}(\boldsymbol{K}_j + \Delta\boldsymbol{K}_j))} = \frac{\exp(\boldsymbol{q}\boldsymbol{K}_i)\exp(\boldsymbol{q}\Delta\boldsymbol{K}_i))}{\sum_{j=1}^{S} \exp(\boldsymbol{q}\boldsymbol{K}_j)\exp(\boldsymbol{q}\Delta\boldsymbol{K}_j)} = \frac{\exp(\boldsymbol{q}\boldsymbol{K}_i))}{\sum_{j=1}^{S} \exp(\boldsymbol{q}\boldsymbol{K}_j)\frac{\exp(\boldsymbol{q}\Delta\boldsymbol{K}_j)}{\exp(\boldsymbol{q}\Delta\boldsymbol{K}_i)}}. \tag{5}$$

If the key quantization error vector $\Delta\boldsymbol{K}_j$ with low quantization precision $B$ is noticeable, the inner product of query and error vector $\boldsymbol{q}\Delta\boldsymbol{K}_j$ is also not ignorable. There are two cases where $\hat{a}_i$ equals to $a_i$ for all tokens. In other words, the attention distribution before and after key quantization are identical.

Case 1) $\frac{\exp(\boldsymbol{q}\Delta\boldsymbol{K}_j)}{\exp(\boldsymbol{q}\Delta\boldsymbol{K}_i)} = 1$, where each key token quantization errors have the same inner product result with the query token $\boldsymbol{q}\Delta\boldsymbol{K}_i = \boldsymbol{q}\Delta\boldsymbol{K}_j$ which normally does not happen.

Case 2) There is a dominating key token $i$. If $j \neq i$, $\exp(\boldsymbol{q}\boldsymbol{K}_i) \gg \exp(\boldsymbol{q}\boldsymbol{K}_j)$ and $\frac{\exp(\boldsymbol{q}\boldsymbol{K}_j)}{\exp(\boldsymbol{q}\boldsymbol{K}_i)} \approx 0$, then

$$\hat{a}_i = \frac{\exp(\boldsymbol{q}\Delta\boldsymbol{K}_i))}{\sum_{j=1}^{S} \exp(\boldsymbol{q}\Delta\boldsymbol{K}_j)\frac{\exp(\boldsymbol{q}\boldsymbol{K}_j)}{\exp(\boldsymbol{q}\boldsymbol{K}_i)}} \approx \frac{\exp(\boldsymbol{q}\Delta\boldsymbol{K}_i))}{\exp(\boldsymbol{q}\Delta\boldsymbol{K}_i))} = 1. \tag{6}$$

Other dominated key token thus has the attention score $\hat{a}_j = 0$. The exactly identical attention distribution with a dominating key token may be a special case, but it indicates that attention heads with a small amount of dominated key tokens, which have highly attention scores and result in sparse and concentrated attention patterns, are consistently robust to low-precision KV cache quantization. $\square$

## B. Effects of KV Cache Quantization Mode and Precision

In this section, we analyze the effects of KV cache quantization mode and precision. We collect the full precision query tensor in the decoding phase and KV cache in both prefilling and decoding stages of the Llama-3.1-8B-Instruct model when processing the first 20 mathematical GSM8K zero-shot prompts without KV cache quantization. After that, we quantize KV cache along the channel or token dimension with uniform precision to compute errors of KV cache and attention score and output vectors of each self-attention layer as defined in Section 3.2, caused by KV cache quantization without any error accumulation. Finally, we average the simulated errors over different prompts and all layers in Table 9 to study the inherent sensitivity of KV cache to quantization mode and precision.

The non-accumulated relative attention output errors $e_o$ of INT8 KV cache quantization with the per-token-asym or per-channel-asym are lower than $3\%$. Minor single-token errors may cause slight shifts in intermediate attention patterns and final output distributions, yet these shifts are typically insufficient to alter the generated output tokens. However, when implementing extremely low-precision 2-bit KV2 cache quantization, the relative key quantization error $e_a$ increases to $40.1\%$ or $77.5\%$, which may lead to substantial attention distribution shift for non-sparse retrieval heads as demonstrated in Figure 4. $e_o$ increases dramatically to $81.4\%$ with the per-channel-asym mode even $96.2\%$ with the per-token-asym mode. The noticeable errors may thus lead to noticeable token flipping and generation errors as in Table 1.

The relative key error $e_k$ of the INT8 per-token-asym key quantization is 0.012280, which is $2.5\times$ larger than the per-channel-asym counterpart 0.004869. Dynamically asymmetric quantization along the channel dimension leads to significantly smaller error of both key cache and attention score compared with token dimension quantization, indicating that key cache is strongly sensitive to quantization dimensions. The phenomenon can be explained with the strong channel-wise outliers of key cache (Liu et al., 2024e; Hooper et al., 2024). While value cache can not benefit from switching the quantization dimension, as the relative value errors of the channel or token dimensions over different precision are quite close.

Table 9: Key and value cache quantization relative error analysis of different precision and quantization methods. We collect BF16 KV cache of 20 prompts from the GSM8K zero-shot dataset with Llama-3.1-8B-Instruct and then perform offline quantization to compute the mean error between BF16 and dequantized KV cache.

| KV cache precision | KV quant mode | Relative $e_k$ | Relative $e_v$ | $e_a$ | Relative $e_o$ |
|---|---|---|---|---|---|
| KV8 | per-channel-asym | 0.004869 | 0.007754 | 0.000013 | 0.027686 |
| | per-token-asym | 0.012280 | 0.007865 | 0.000018 | 0.014589 |
| KV4 | per-channel-asym | 0.080991 | 0.125457 | 0.000172 | 0.158429 |
| | per-token-asym | 0.196476 | 0.126894 | 0.000251 | 0.206909 |
| KV2 | per-channel-asym | 0.401151 | 0.604678 | 0.000868 | 0.814023 |
| | per-token-asym | 0.774668 | 0.607898 | 0.001166 | 0.961792 |

As shown in Figure 7, there are clear layer-wise diversities of KV quantization errors $e_k$ and $e_o$ with different quantization modes including per-token-asym and per-channel-asym and different precision like INT8, INT4, and INT2. In addition, changing the quantization dimension or mode can result in the significant distribution shift of layer-wise key quantization error. For example, the most sensitive layer with the per-token-asym quantization mode is layer-29, while it changes to layer-11 and layer-13 with the per-token-asym mode. Statically retaining the first or last several layers with more sparse budgets (Tang et al., 2024) may not general well in KV cache quantization. Therefore, we need an automatic KV cache quantization tuning framework to adaptively adopt to these layer-wise differences and configuration modifications.

## C. Experimental Settings

KVTuner is an automatic KV cache quantization precision tuning framework and can be applied to any quantization mode. We choose two representative and efficient KV cache quantization algorithms KIVI (Liu et al., 2024e) and per-token-asym with uniform KV8, KV4, or KV2 precision pairs in all layers as baselines. Specifically, for the KIVI quantization method, we set the residual length to 32 and the group size to 32. KVTuner is currently implemented based on huggingface transformers, but it can be applied to inference frameworks such as vLLM (Kwon et al., 2023), Megatron (Shoeybi et al., 2019), LMDeploy (Contributors, 2023), and SGLang (Zheng et al., 2024). To ensure compatibility, we integrate KV cache quantization methods including KIVI, per-token-asym, and KVTuner within the lm-evaluation-harness (Gao et al., 2024), allowing for seamless adaptation and reproducibility of KVTuner.

We select three popular and recently released LLMs series Llama3.1 (Dubey et al., 2024), Mistral-v0.3 (Jiang et al., 2023), and Qwen2.5 (Yang et al., 2024a). Among them, Llama-3.1-8B-Instruct, Mistral-7B-Instruct-v0.3, and Qwen2.5-7B-Instruct represent the most studied model size. To cover more LLMs application scenarios with different scales, Qwen2.5-3B-Instuct and its quantized version Qwen2.5-3B-Instruct-AWQ are selected for personal devices with limited GPU memory, while Qwen2.5-{14B, 32B}-Instuct with larger model scale and better performance are also tested. We also test Qwen2.5-Math-7B-Instruct for mathematical reasoning tasks.

We cover 5 general AIGC and 2 mathematical reasoning tasks available in lm-evaluation-harness . 1) **General tasks**: CEVAL(Huang et al., 2023), MMLU (Hendrycks et al., 2021), TriviaQA (Joshi et al., 2017), RACE (Lai et al., 2017), and TruthfulQA (Lin et al., 2022). 2) **Math, science, and logic tasks**: GSM8K {0-shot, 4-shot, 8-shot, 16-shot} (Cobbe et al., 2021), GSM8K multi-round with lm-evaluation-harness (Gao et al., 2024), GPQA (Rein et al., 2023).

For the final layer-wise KV cache quantization precision pair searching with multi-objective optimization, we use the open-sourced and widely used Optuna framework (Akiba et al., 2019) and MOEA/D (Zhang & Li, 2007) algorithm. In which case, we treat the LLM inference accuracy under different layer-wise KV precision pairs and input prompts as block-box. The intra-layer and inter-layer search space pruning only takes several minutes but significantly improves sampling efficiency of the downstream Optuna.

We first preprocess the available quantization precision options for each layer group and store them in an array. The indices of this array are then treated as integer parameters, which are optimized by Optuna through multi-objective optimization. The first objective is to maximize the accuracy on the first 200 samples of the GSM8K dataset, while the second objective is to minimize the equivalent quantization precision or memory usage of KV cache. For each combination of model and quantization mode, we set a soft constraint on the equivalent precision at 4-bit and 6-bit for optuna, conducting 200 search iterations for each setting. The total time cost of offline KV cache precision pair tuning with Optuna mainly depends on the

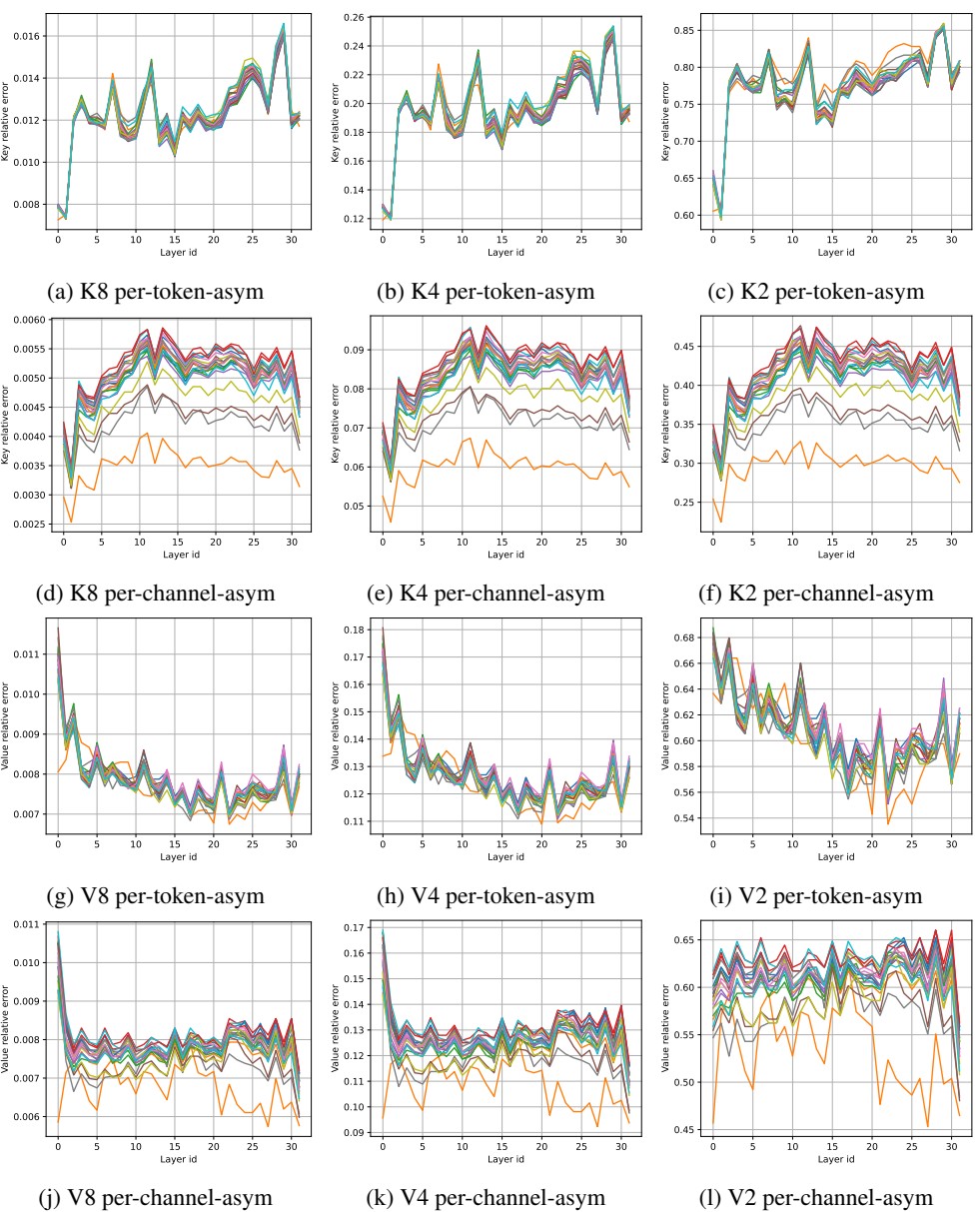

(a) K8 per-token-asym     (b) K4 per-token-asym     (c) K2 per-token-asym

(d) K8 per-channel-asym     (e) K4 per-channel-asym     (f) K2 per-channel-asym

(g) V8 per-token-asym     (h) V4 per-token-asym     (i) V2 per-token-asym

(j) V8 per-channel-asym     (k) V4 per-channel-asym     (l) V2 per-channel-asym

Figure 7: Layer-wise relative key errors $e_k$ and value errors $e_v$ of **Llama-3.1-8B-Instruct** with the **per-token-asym and per-channel-asym quantization** modes, KV cache precision as 8, 4, and 2-bit, and the same settings in Table 9.

hardware and operator implementation efficiency.

# D. Search Space Pruning and Multi-objective Optimization results

## D.1. Intra-Layer and Inter-Layer Search Space Pruning Results

### D.1.1. INTRA-LAYER PARETO OPTIMAL KV CACHE PRECISION PAIR PRUNING

The intra-layer KV cache quantization precision pair pruning based on Pareto frontier are available in Table 4. The calibration dataset is the first 20 prompts from the zeroshot GSM8K dataset. The Pareto optimal KV cache precision pairs in most layers are the key-first set {KV8, K8V4, KV4, K4V2, KV2}, indicating that the observation that key cache is more important than value cache holds.

When both key and value cache are quantized along the token dimension, only the first layer in Llama-3.1-8B-Instruct and Mistral-7B-Instruct-v0.3 prefers other KV precision pairs and all layers in Qwen2.5-{14B, 32B}-Instruct select the key-first set. In contrast, K8V2 outperforms KV4 in four important layers of Qwen2.5-{3B, 7B}-Instruct, indicating that uniform 4-bit key quantization may lead to model accuracy degradation as in Table 13.

When utilizing the KIVI-like key per-channel-asym and value per-token-asym quantization mode, more layers show diverse preference of Pareto optimal KV cache quantization precision pairs. In these layers, K4V8 and K2V4 outperform K8V4 and K4V2, which means that lower precision key is better than lower precision value in terms of attention errors. It indicates that per-channel key quantization can effectively reduce quantization errors.

### D.1.2. INTER-LAYER CLUSTERING BASED ON ATTENTION ERRORS

After the intra-layer KV cache quantization precision pair pruning, we apply the inter-layer clustering among the layers with the same precision pair set. The clustering algorithm is DBSCAN (Ester et al., 1996) with the hyper-parameter epsilon=0.05 and min_samples=2. As demonstrated in Table 10, we successfully reduce the exponential component of search space size from the number of transformer layers $L$ e.g. 28~64 to the number of clustered layer groups $G$ e.g. 4~8. Utilizing the two-level search space pruning, the total number of combinations of candidate KV cache precision pairs is significantly reduced from $9^L$ to $5^G$ or $6^G$. In Llama-3.1-8B-Instruct, $9^L = 9^{32} \approx 3.4 \times 10^{30}$, while $5^G = 5^6 = 15625$.

In the layer-wise relative attention output errors with per-token-asym KV cache quantization of Llama-3.1-8B-Instruct in Figure 13, the highly sensitive layers include layer-{0, 1, 2, 3, 4, 23, 24, 25, 27, 28, 29}, while the insensitive layers include layer-{8, 9, 10, 11, 13, 14, 15, 20, 30}. Layers in these two classes are correctly clustered into different groups. Similar phenomenon can also be observed in Qwen2.5-7B-Instruct per-token-asym and KIVI-like quantization modes in Figure 16 and 18, respectively. Therefore, we can conclude that the proposed multi-objective Pareto frontier based intra-layer pruning and inter-layer clustering algorithms successfully reduce the search space by considering the inherent layer-wise sensitivities.

## D.2. Searched layer-wise KV precision pairs

The final searched layer-wise mixed precision KV cache quantization precision pairs of different LLMs and KV quantization modes are available in Table 11. Some clustered layer groups in Table 10 choose the same KV cache quantization pairs under the given memory consumption and/or accuracy degradation constraints. The number of utilized KV cache quantization precision pairs is reduced from $6 \sim 8$ to $2 \sim 5$ in the tested Llama-3.1-8B-Instruct, Qwen2.5-3B-Instruct, and Qwen2.5-7B-Instruct models. In addition, the significantly diverse layer-wise KV precision pair distribution in Table 11 indicates that there are not clear heuristic rules based on layer depths to identify layer importance and sensitivity to KV cache quantization. Therefore, we need to measure the model accuracies considering their complicated nonlinear dependencies to layer-wise KV cache precision pairs and utilize accuracies to distinguish the whole model level KV cache precision pair combinations.

## D.3. Pareto Frontier with the GSM8K Calibration Dataset

We use the open-sourced package optuna (Akiba et al., 2019) with the MOEA/D algorithm (Zhang & Li, 2007) for the final search with the model memory usage and inference accuracy of the first 200 4-shot GSM8K prompts. The multi-objective search of the Pareto optimal layer-wise KV cache quantization precision pairs of Llama-3.1-8B-Instruct, Mistral-7B-Instruct-v0.3, Qwen2.5-3B-Instruct, and Qwen2.5-7B-Instruct with the KIVI and per-token-asym quantization modes are available

Table 10: Inter-layer clustering results by clustering among the layers with the same pruned intra-layer KV cache quantization precision pairs. For example, layers 14 and 20 demonstrate higher sensitivity than layers 3, 13, and 27 as visualized in Figure 16. They are clustered into different group, validating the effectiveness of our intra-layer pruning and inter-layer clustering.

| Model name | L | Key quant. mode | G | Grouped layer ids |
|---|---|---|---|---|
| Llama-3.1-8B-Instruct | 32 | per-token-asym | 6 | {0}, {1∼4, 7, 13, 18, 25, 27, 31}, {5, 6, 12, 21, 26, 28}, {8∼11, 14∼17, 20, 30}, {19, 22}, {23, 24, 29} |
| | | per-channel-asym | 6 | {0}, {1∼3, 7, 29, 31}, {4, 25, 27}, {5, 21, 23, 24}, {6, 8∼12, 14∼16, 18∼20, 22, 26, 28, 30}, {13, 17} |
| Mistral-7B-Instruct-v0.3 | 32 | per-token-asym | 5 | {0}, {1, 2}, {3, 4, 23, 31}, {5, 6}, {7∼22, 24∼30} |
| | | per-channel-asym | 8 | {0, 1, 31}, {2∼4}, {6, 27, 29}, {7, 8, 10, 18}, {9, 14}, {5, 21∼26, 28, 30}, {11∼13, 15, 17, 19, 20}, {16} |
| Qwen2.5-3B-Instruct | 36 | per-token-asym | 8 | {0}, {1, 3∼6, 8, 9, 12, 13, 15, 20}, {2, 14, 23, 35}, {7, 11, 16, 25, 28, 32}, {10, 19, 24, 26, 33}, {17, 30, 31, 34}, {21, 22}, {18, 27, 29} |
| | | per-channel-asym | 8 | {0, 1}, {2, 4}, {34, 35}, {3, 6, 11, 13, 23}, {5, 7, 25, 32, 33}, {8, 16, 18, 21, 22, 24, 26, 27, 30}, {9, 10, 14, 15, 17, 19, 20, 29, 31}, {12, 28} |
| Qwen2.5-7B-Instruct | 28 | per-token-asym | 8 | {0}, {1, 2, 4, 5, 25}, {6, 19}, {7, 10, 11, 15, 23}, {8, 24}, {9, 12, 16∼18, 21, 22, 26}, {14, 20}, {3, 13, 27} |
| | | per-channel-asym | 7 | {0, 2}, {1, 3}, {4, 5, 12, 22∼25}, {7, 9, 10, 13, 14, 16, 18∼21, 27}, {8, 26}, {11, 15, 17}, {6} |
| Qwen2.5-14B-Instruct | 48 | per-token-asym | 6 | {0∼2, 6, 11, 12, 19, 23∼25, 41}, {3∼5, 8}, {7, 10, 15}, {9, 13, 14, 31, 38, 39}, {16∼18, 20, 21, 27, 28, 30, 32∼37, 40, 42∼44, 46, 47}, {22, 26, 29, 45} |
| | | per-channel-asym | 7 | {0, 2}, {1, 3, 4}, {5, 6, 8, 9, 12}, {7, 10, 13, 15∼21, 23, 24, 26∼33, 35∼38, 44∼47}, {11, 25, 41, 42}, {14, 39, 40, 43}, {22, 34} |
| Qwen2.5-32B-Instruct | 64 | per-token-asym | 4 | {0, 2, 11, 12, 15, 33, 54, 57}, {1, 5, 7∼10, 13, 14, 17∼32, 34∼53, 55, 56, 58∼63}, {3, 4}, {6, 16} |
| | | per-channel-asym | 5 | {0∼4}, {11}, {5∼10, 12, 14, 16, 18∼23, 26∼28, 32}, {13, 15, 17, 22, 24, 25, 29∼31, 33∼62}, {63} |

in Figure 8 and 9. In order to validate the effectiveness of the proposed two-stage intra-layer and inter-layer search space pruning, we disable the pre-processing process and directly use the original full $S^L$ search space in Figure 10.

For each combination of model and quantization mode, we set a soft constraint on the equivalent precision at 4-bit and 6-bit for optuna, conducting 200 search iterations for each setting. The search results are then merged for visualization. In cases where the two-stage intra-layer and inter-layer search space pruning is not applied, we set the maximum equivalent quantization precision to 6-bit and similarly perform 200 search iterations. Specifically, for the KIVI quantization method, we set the residual length to 32 and the group size to 32.

Note that for Qwen-2.5-7B with the KIVI quantization mode, the result from 200 search iterations appeared abnormal. Therefore, we extended the search to 500 iterations to obtain the final result.

## E. Correlation of Model- and Layer-wise KV Cache Quantization Sensitivity with Attention Patterns

According to the layer-wise attention score errors of Llama-3.1-8B-Instruct in Figure 3 and Qwen2.5-7B-Instruct in Figure 16, we can observe the clear layer-wise difference in the same LLM. In this section, we try to explain the reason of the difference from the attention pattern perspective as in Figure 11 and 12. In which, we visualize block level attention scores of the first 4 heads with block size 4 in the prefilling and decoding stages, and horizontal and vertical axes represent the key and query dimensions respectively. Yellow, green, and purple points indicate high, medium, and low attention scores, respectively. We find out that the more complex and dynamic attention patterns usually lead to larger attention score errors and sensitivity to KV cache quantization of intermediate transformer layers and the whole LLMs.

Take Llama-3.1-8B-Instruct as an example, layer 12 and 13 are in the group with high attention score errors, while layer 0 and 31 are in the medium error group and layer 2 and 23 are in the low error group. Analyzing the attention patterns of these layer in the below Figure 11, we can conclude that heads in the layer 12 and 13 have dynamic and non-sparse attention patterns, which are called as retrieval heads (Tang et al., 2025; Xiao et al., 2025). In contrast, heads in layer 0, 2, 23 and 31 have more static attention patterns like attention sink and recent window, which are called as streaming heads (Xiao et al., 2025; 2024).

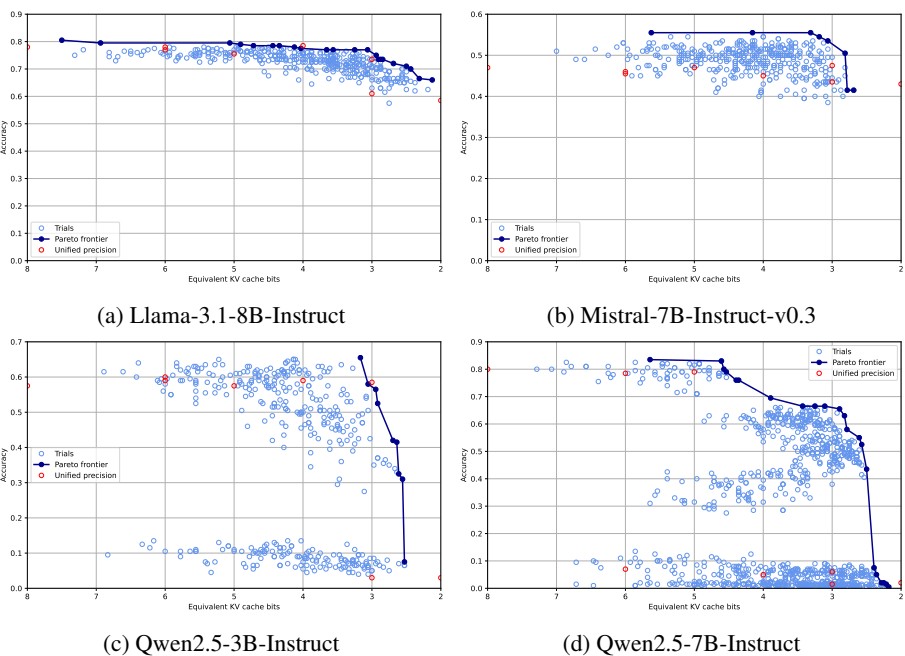

(a) Llama-3.1-8B-Instruct

(b) Mistral-7B-Instruct-v0.3

(c) Qwen2.5-3B-Instruct

(d) Qwen2.5-7B-Instruct

Figure 8: Pareto frontier of different models with the **KIVI** quantization mode on the first 200 data slices of the 4-shot GSM8K dataset.

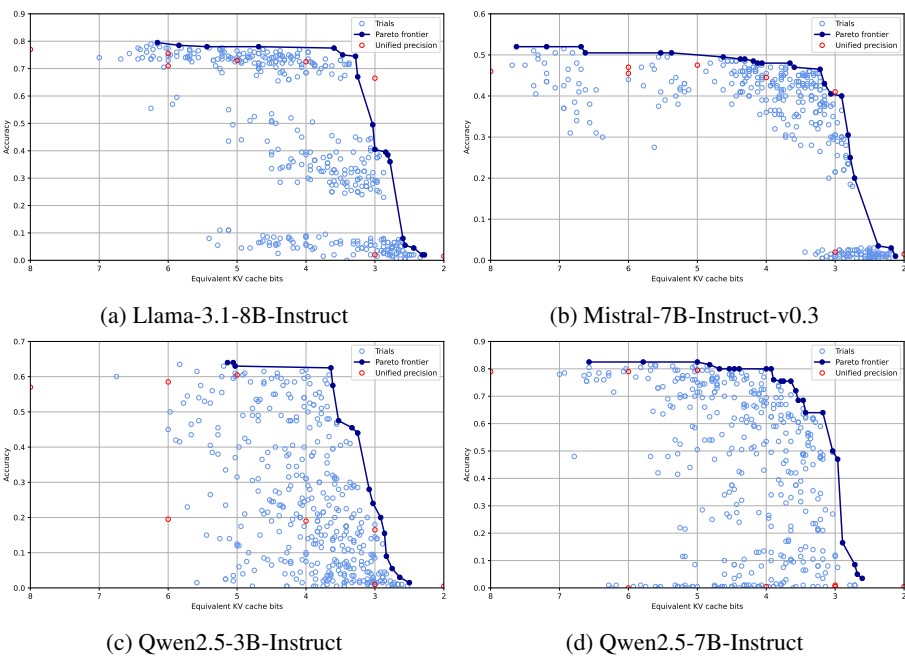

(a) Llama-3.1-8B-Instruct

(b) Mistral-7B-Instruct-v0.3

(c) Qwen2.5-3B-Instruct

(d) Qwen2.5-7B-Instruct

Figure 9: Pareto frontier of different models with the **per-token-asym** quantization mode on the first 200 data slices of the 4-shot GSM8K dataset.

Table 11: Detailed searched layer-wise KV cache quantization precision pairs of different LLMs and KV cache quantization modes by KVTuner.

| Model name | Quant. mode | Equivalent precision | Quant. precision | Layer ids |
|---|---|---|---|---|
| Llama-3.1-8B-Instruct | per-token-asym | 3.59 | K4V8 | 0 |
| | | | KV4 | 5, 6, 8–12, 14–17, 20, 21, 26, 28, 30 |
| | | | K4V2 | 1–4, 7, 13, 18, 19, 22–25, 27, 29, 31 |
| | | 5.44 | K8V4 | 1–4, 7–11, 13–18, 20, 23–25, 27, 29–31 |
| | | | KV4 | 0, 5, 6, 12, 19, 21, 22, 26, 28 |
| | KIVI | 3.25 | K8V4 | 13, 17 |
| | | | KV4 | 1–3, 7, 29, 31 |
| | | | K4V2 | 5, 6, 8–12, 14–16, 18–24, 26, 28, 30 |
| | | | KV2 | 0, 4, 25, 27 |
| | | 4.90 | K8V4 | 4, 6, 8–12, 14–16, 18–20, 22, 25–28, 30 |
| | | | KV4 | 1–3, 7, 29, 31 |
| | | | K4V2 | 5, 21, 23, 24 |
| | | | K2V4 | 0 |
| | | | KV2 | 13, 17 |
| Qwen2.5-3B-Instruct | per-token-asym | 4.00 | K8V4 | 17, 18, 27, 29–31, 34 |
| | | | K8V2 | 0 |
| | | | KV4 | 7, 10, 11, 19, 21, 22, 24–26, 28, 32, 33 |
| | | | K4V2 | 1–6, 8, 9, 12, 13–16, 20, 23, 35 |
| | | 5.06 | KV8 | 0 |
| | | | K8V4 | 1, 3–6, 8–10, 12, 13, 15, 17–20, 24, 26, 27, 29–31, 33, 34 |
| | | | K4V2 | 2, 7, 11, 14, 16, 21–23, 25, 28, 32, 35 |
| | KIVI | 3.17 | K4V8 | 0–1 |
| | | | K4V2 | 3, 5–11, 13–27, 29–33 |
| | | | KV4 | 34–35 |
| | | | K2V4 | 2, 4 |
| | | | KV2 | 12, 28 |
| | | 3.44 | KV8 | 0–1 |
| | | | KV4 | 3, 5–7, 11, 13, 23, 25, 32, 33 |
| | | | K4V2 | 8–10, 14–22, 24, 26, 27, 29–31 |
| | | | K2V4 | 34–35 |
| | | | KV2 | 2, 4, 12, 28 |
| Qwen2.5-7B-Instruct | per-token-asym | 4.00 | KV8 | 0 |
| | | | K8V2 | 3, 13, 27 |
| | | | KV4 | 6, 7, 9–12, 14–23, 26 |
| | | | K4V2 | 1, 2, 4, 5, 8, 24, 25 |
| | | 5.00 | K8V4 | 8, 9, 12, 14, 16–18, 20–22, 24, 26 |
| | | | K8V2 | 0, 3, 13, 27 |
| | | | KV4 | 1, 2, 4–7, 10, 11, 15, 19, 23, 25 |
| | KIVI | 3.92 | KV8 | 0, 2, 6, 11, 15, 17 |
| | | | KV4 | 4, 5, 8, 12, 22–26 |
| | | | KV2 | 1, 3, 7, 9, 10, 13, 14, 16, 18–21, 27 |
| | | 5.96 | KV8 | 0, 2, 7, 9, 10, 13, 14, 16, 18–21, 27 |
| | | | K8V4 | 4, 5, 12, 22, 23, 24, 25 |
| | | | K4V2 | 11, 15, 17 |
| | | | K2V4 | 1, 3 |
| | | | KV2 | 6, 8, 26 |

Compared with Llama-3.1-8B-Instruct which has the high ratio of heads with static attention patterns, Qwen2.5-7B-Instruct consists of many heads with mixture of dynamic retrieval heads and other static patterns. It may explain why Qwen2.5-7B-Instruct is more unstable to KV cache quantization as in Table 2. Layer 5, 12, 21, and 27 have similar attention patterns, but the relative strength of retrieval and streaming heads leads to the difference of layer-wise sensitivity to KV cache quantization.

However, the sensitivity to KV cache quantization is the inherent model property which can be learned offline. Therefore, it is necessary to apply layer-wise mixed precision KV cache quantization and maintain high precision of key cache than value cache with multi-objective optimization KV precision pair tuning as proposed in this work. KVTuner thus makes equivalent 4-bit and even lower KV cache quantization nearly lossless in the sensitive models like Qwen2.5-7B-Instruct.

### E.1. More KV Cache Quantization Results on General and Mathematical Reasoning Datasets

The experimental results of the selected 5 LLMs on the general and mathematical reasoning datasets with uniform KV cache quantization precision pairs are available in Table 12 and 13. To simulate the Openai o1 like long CoT reasoning process, the few-shot CoTs in the GSM8K dataset are treated as a multi-turn conversation, which is enabled with the flags

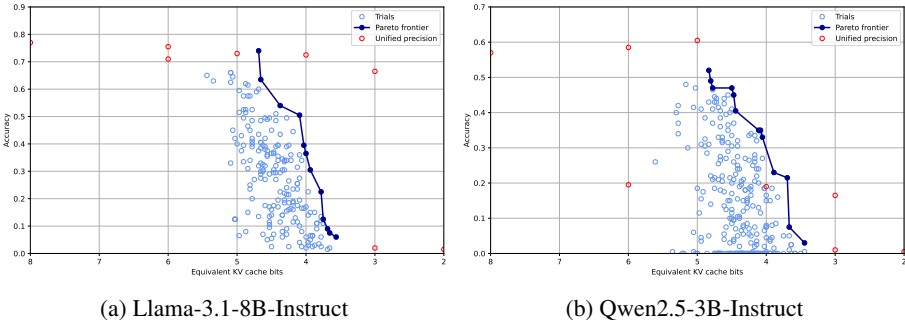

    (a) Llama-3.1-8B-Instruct                (b) Qwen2.5-3B-Instruct

Figure 10: Pareto frontier of different models with the **per-token-asym** quantization model on the first 200 data slices of the 4-shot GSM8K dataset without intra-layer and inter-layer search space pruning.

*fewshot_as_multiturn* and *apply_chat_template* in lm-evaluation-harness. In which cases, questions are provided as user content and answers are provided as assistant responses instead of directly using the given standard answers. Table 14 summarizes the results of 8 LLMs including Qwen2.5-{14, 32B}-Instruct under the *fewshot_as_multiturn* setting.

There are limited long output mathematical reasoning datasets in lm-evaluation-harness (Gao et al., 2024) and the evaluation of the long context generation is an open question. Therefore, we enable KV cache quantization in both the prefilling and decoding stages to amplify the effects to final generation results caused by KV cache quantization error accumulation, which makes distinguishing different quantization methods easier. For the KIVI quantization mode, we utilize the HQQ quantizer from HuggingFace's implementation, with both the residual length and group size set to 32.

According to Table 12, 13, and 14, most LLMs, including Llama-3.1-8B-Instruct, Mistral-7B-Instruct-v0.3, Qwen2.5-{14B, 32B}-Instruct, are robust to low-bit KV cache quantization. Although error accumulation caused by KV cache quantization starts from the prefilling stage, the high KV cache quantization precision pair KV8 with KIVI or per-token-asym quantization mode are still generally lossless, except Qwen2.5-Math-7B-Instruct. The uniform KV cache quantization precision pairs KV4 or even K4V2 with the KIVI quantization mode can achieve nearly lossless $4\times$ or even $5.3\times$ KV cache compression, respectively. KV4 with the simple per-token-asym mode also results in negligible accuracy loss in Llama-3.1-8B-Instruct and Mistral-7B-Instruct-v0.3 as shown in Table 12. KIVI does outperform the per-token-asym quantization mode in the low-precision settings such as KV4, K4V2, and KV2, especially in Qwen2.5-3B-Instruct-AWQ and Qwen2.5-7B-Instruct as demonstrated in Figure 13.

As shown in Figure 14, the larger Qwen2.5-{14B,32B}-Instruct models are robust than the smaller Qwen2.5-{3B, 7B, Math-7B}-Instruct and the weight quantized Qwen2.5-3B-Instruct-AWQ models. In addition, comparing Qwen2.5-3B-Instruct-AWQ and Qwen2.5-3B-Instruct, we can conclude that model weight quantization with AWQ does not affect the model-level sensitivity to KV cache quantization. The increasing GSM8K accuracy with the longer CoTs under the half precision BF16 KV cache setting indicates that most Qwen2.5 models benefit from longer CoTs. We also obverse that 16-shot CoTs with K4V2 KV cache precision outperforms the 4-shot CoTs with BF16 KV cache precision on the larger Qwen2.5-{14B,32B}-Instruct models. It indicates that longer CoT with lower and mixed precision KV cache outperforms uniform precision counterparts as in Section 6.2. In other words, mixed precision key cache quantization with higher precision key can achieve both memory usage and inference accuracy improvement than equal precision key and value cache quantization.

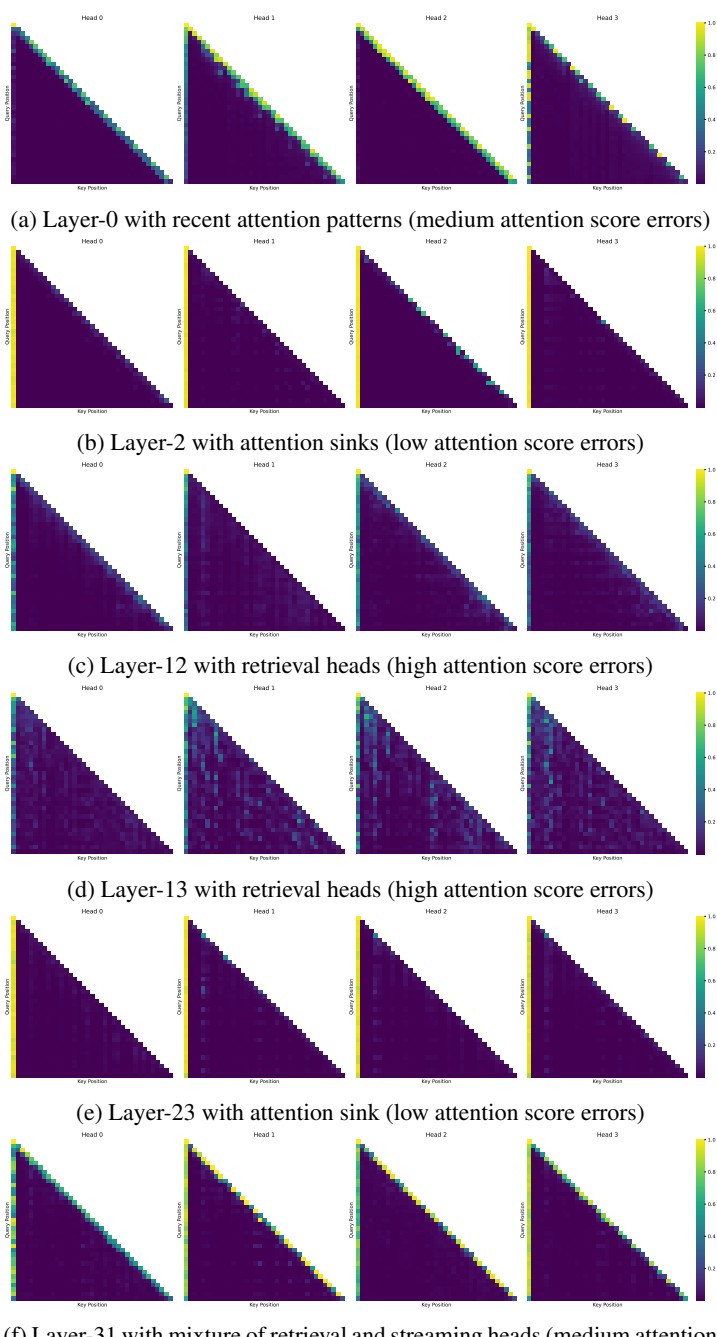

(a) Layer-0 with recent attention patterns (medium attention score errors)

(b) Layer-2 with attention sinks (low attention score errors)

(c) Layer-12 with retrieval heads (high attention score errors)

(d) Layer-13 with retrieval heads (high attention score errors)

(e) Layer-23 with attention sink (low attention score errors)

(f) Layer-31 with mixture of retrieval and streaming heads (medium attention score errors)

Figure 11: Selected layer-wise attention patterns of **Llama-3.1-8B-Instruct** model and the first prompt in the **0-shot GSM8K** dataset. Many layers and heads of Llama-3.1-8B-Instruct have simple and streaming attention patterns which highly concentrated and sparse attention scores. As a result, the attention score errors in these layers are medium or low. In contrast, layers with retrieval or mixed attention patterns, whose attention scores are non-sparse, normally show high attention score errors. *We also observe that the attention patterns of query heads in the same group and sharing the same key cache are highly similar, which may indicate that we can apply attention head group-wise KV cache management for better accuracy.*

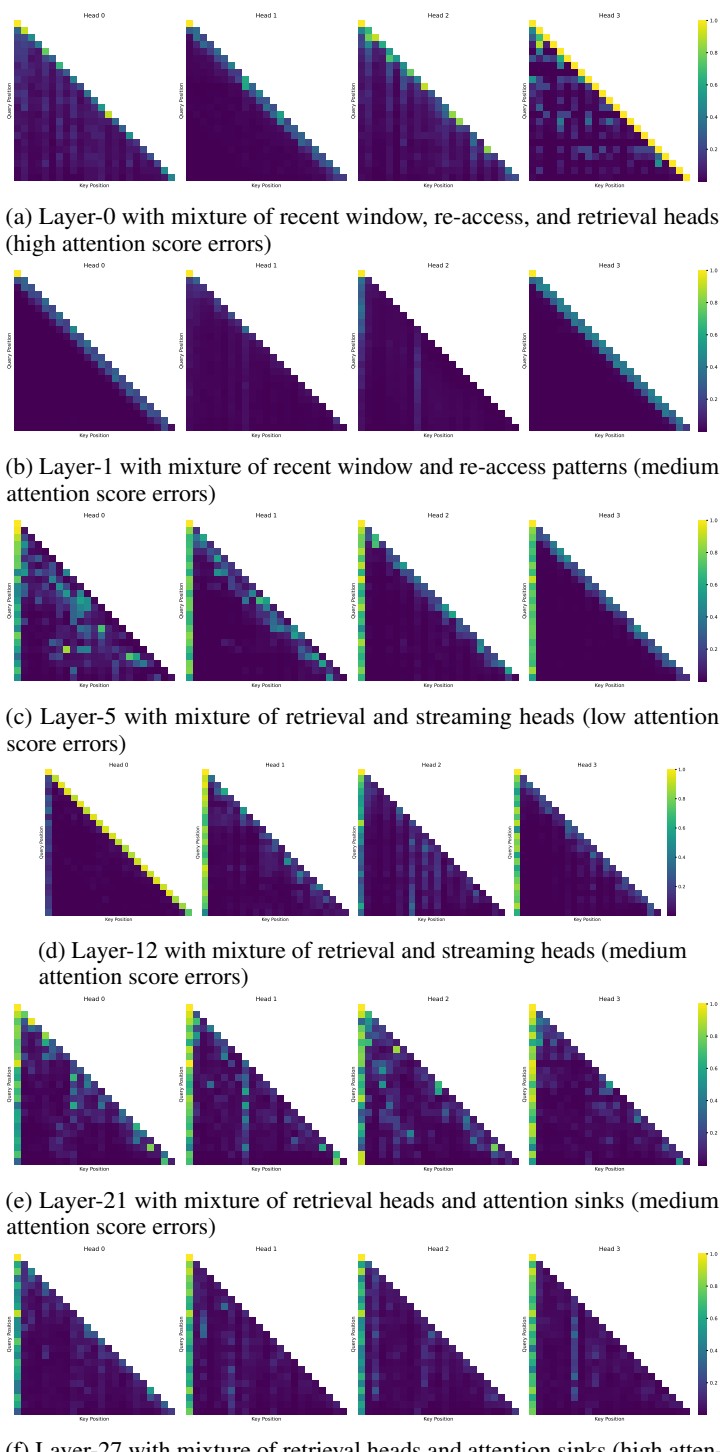

(a) Layer-0 with mixture of recent window, re-access, and retrieval heads (high attention score errors)

(b) Layer-1 with mixture of recent window and re-access patterns (medium attention score errors)

(c) Layer-5 with mixture of retrieval and streaming heads (low attention score errors)

(d) Layer-12 with mixture of retrieval and streaming heads (medium attention score errors)

(e) Layer-21 with mixture of retrieval heads and attention sinks (medium attention score errors)

(f) Layer-27 with mixture of retrieval heads and attention sinks (high attention score errors)

Figure 12: Selected layer-wise attention patterns of **Qwen2.5-7B-Instruct** model and the first prompt in the **0-shot GSM8K** dataset. Most layers and heads of Qwen2.5-7B-Instruct have complex attention patterns, such as retrieval, and mixture of retrieval and recent or attention sink patterns. These non-sparse and non-concentrated attention patterns result in the high sensitivity of Qwen2.5-7B-Instruct to KV cache compression including low-precision quantization and even model weight and activation quantization.

Table 12: Final generation accuracy comparison of different KV cache quantization modes and precisions and Llama-3.1-8B-Instruct and Mistral-7B-Instruct-v0.3 on the AIGC and mathematical datasets. KV cache quantization is enabled during both prefilling and decoding stages to amplify the effects of error accumulation.

| Quant. method | Precision | CEVAL | MMLU | TriviaQA | RACE | TruthfulQA | GSM8K | | | | Average |
| | | | | | | | 0-shot | 4-shot | 8-shot | 16-shot | |
|---|---|---|---|---|---|---|---|---|---|---|---|
| | | | | | **Llama-3.1-8B-Instruct** | | | | | | |
| BF16 | BF16 | 0.5386 | 0.6802 | 0.5161 | 0.4469 | 0.6267 | 0.2866 | 0.7635 | 0.7741 | 0.7854 | 0.6020 |
| KIVI | KV8 | 0.5416 | 0.6798 | 0.5162 | 0.4469 | 0.6304 | 0.2752 | 0.7597 | 0.7657 | 0.7809 | 0.5996 |
| | K8V4 | 0.5394 | 0.6792 | 0.5138 | 0.4498 | 0.6450 | 0.2858 | 0.7695 | 0.7794 | 0.7923 | 0.6060 |
| | K8V2 | 0.4807 | 0.6381 | 0.4989 | 0.4383 | 0.6499 | 0.2358 | 0.7074 | 0.7036 | 0.7195 | 0.5636 |
| | K4V8 | 0.5327 | 0.6694 | 0.5144 | 0.4488 | 0.5851 | 0.2623 | 0.7566 | 0.7566 | 0.7710 | 0.5885 |
| | KV4 | 0.5245 | 0.6689 | 0.5135 | 0.4498 | 0.6132 | 0.2782 | 0.746 | 0.7589 | 0.7680 | 0.5912 |
| | K4V2 | 0.4703 | 0.6236 | 0.5016 | 0.4450 | 0.5373 | 0.2464 | 0.6694 | 0.6694 | 0.6854 | 0.5387 |
| | K2V4 | 0.3247 | 0.4628 | 0.4761 | 0.3675 | 0.4639 | 0.0978 | 0.1122 | 0.1054 | 0.0842 | 0.2772 |
| | KV2 | 0.2771 | 0.3600 | 0.4584 | 0.3301 | 0.3182 | 0.0508 | 0.0432 | 0.0318 | 0.0250 | 0.2105 |
| Per-token-asym | KV8 | 0.5342 | 0.6800 | 0.5175 | 0.4459 | 0.6206 | 0.2805 | 0.7657 | 0.7809 | 0.7801 | 0.6006 |
| | K8V4 | 0.5386 | 0.6776 | 0.4709 | 0.4450 | 0.6169 | 0.3154 | 0.7733 | 0.7688 | 0.7847 | 0.5990 |
| | K8V2 | 0.4792 | 0.6183 | 0.4984 | 0.4239 | 0.5887 | 0.1501 | 0.6391 | 0.6262 | 0.6550 | 0.5199 |
| | K4V8 | 0.5163 | 0.6579 | 0.5123 | 0.4411 | 0.6781 | 0.2517 | 0.7180 | 0.7293 | 0.7240 | 0.5810 |
| | KV4 | 0.5141 | 0.6570 | 0.4849 | 0.4325 | 0.6340 | 0.2782 | 0.7240 | 0.7202 | 0.7157 | 0.5734 |
| | K4V2 | 0.4413 | 0.5910 | 0.4779 | 0.4306 | 0.5447 | 0.1289 | 0.5709 | 0.5633 | 0.5519 | 0.4778 |
| | K2V4 | 0.2400 | 0.2350 | 0.0249 | 0.2593 | 0.3268 | 0.0212 | 0.0159 | 0.0296 | 0.0212 | 0.1304 |
| | KV2 | 0.2444 | 0.2338 | 0.0052 | 0.2478 | 0.2277 | 0.0227 | 0.0174 | 0.0197 | 0.0273 | 0.1162 |
| | | | | | **Mistral-7B-v0.3** | | | | | | |
| BF16 | BF16 | 0.3923 | 0.5911 | 0.6081 | 0.4057 | 0.4296 | 0.0766 | 0.3389 | 0.3753 | 0.3601 | 0.3975 |
| KIVI | KV8 | 0.3945 | 0.5901 | 0.6072 | 0.4115 | 0.4259 | 0.0735 | 0.3412 | 0.3639 | 0.3624 | 0.3967 |
| | K8V4 | 0.3945 | 0.5909 | 0.6068 | 0.4067 | 0.4394 | 0.0781 | 0.3457 | 0.3723 | 0.3669 | 0.4001 |
| | K8V2 | 0.3819 | 0.5776 | 0.6042 | 0.4086 | 0.4370 | 0.0675 | 0.3404 | 0.3518 | 0.3609 | 0.3922 |
| | K4V8 | 0.3990 | 0.5875 | 0.6069 | 0.4048 | 0.4308 | 0.0697 | 0.3442 | 0.3563 | 0.3738 | 0.3970 |
| | KV4 | 0.3945 | 0.5886 | 0.6074 | 0.4105 | 0.4455 | 0.0751 | 0.3434 | 0.3662 | 0.3586 | 0.3989 |
| | K4V2 | 0.3752 | 0.5753 | 0.6035 | 0.4000 | 0.4223 | 0.0705 | 0.3434 | 0.3397 | 0.3616 | 0.3879 |
| | K2V4 | 0.3128 | 0.4926 | 0.5982 | 0.3847 | 0.3917 | 0.0637 | 0.0978 | 0.0910 | 0.0773 | 0.2789 |
| | KV2 | 0.2905 | 0.4571 | 0.5920 | 0.3885 | 0.4688 | 0.0478 | 0.0766 | 0.0644 | 0.0516 | 0.2708 |
| Per-token-asym | KV8 | 0.3900 | 0.5892 | 0.6071 | 0.4067 | 0.4284 | 0.072 | 0.3419 | 0.3745 | 0.3571 | 0.3963 |
| | K8V4 | 0.3967 | 0.5897 | 0.6040 | 0.4057 | 0.4357 | 0.0751 | 0.3533 | 0.3715 | 0.3707 | 0.4003 |
| | K8V2 | 0.3692 | 0.5760 | 0.5797 | 0.4029 | 0.3929 | 0.0675 | 0.3328 | 0.3381 | 0.3548 | 0.3793 |
| | K4V8 | 0.3871 | 0.5862 | 0.6070 | 0.4077 | 0.4259 | 0.0629 | 0.3450 | 0.3578 | 0.3692 | 0.3943 |
| | KV4 | 0.3871 | 0.5865 | 0.5994 | 0.4048 | 0.4321 | 0.072 | 0.3450 | 0.3556 | 0.3685 | 0.3946 |
| | K4V2 | 0.3618 | 0.5672 | 0.5774 | 0.4086 | 0.3623 | 0.0599 | 0.3048 | 0.3389 | 0.3571 | 0.3709 |
| | K2V4 | 0.2786 | 0.4360 | 0.4688 | 0.3914 | 0.3268 | 0.0303 | 0.0334 | 0.0281 | 0.0212 | 0.2238 |
| | KV2 | 0.2741 | 0.3926 | 0.4045 | 0.4019 | 0.2999 | 0.0281 | 0.0265 | 0.0167 | 0.0220 | 0.2074 |

Table 13: Final generation accuracy comparison of different KV cache quantization modes and precisions and Qwen2.5 LLMs on the AIGC and mathematical datasets. KV cache quantization is enabled during both prefilling and decoding stages to amplify the effects of error accumulation.

| Quant. method | Precision | CEVAL | MMLU | TriviaQA | RACE | TruthfulQA | GSM8K 0-shot | 4-shot | 8-shot | 16-shot | Average |
|---|---|---|---|---|---|---|---|---|---|---|---|
| | | | | | | | | | | | |
| BF16 | BF16 | 0.7125 | 0.6382 | 0.2299 | 0.3904 | 0.4700 | 0.4867 | 0.5815 | 0.6353 | 0.6861 | 0.5367 |
| KIVI | KV8 | 0.7073 | 0.6389 | 0.2283 | 0.3885 | 0.4761 | 0.4882 | 0.5762 | 0.6361 | 0.6816 | 0.5357 |
| | K8V4 | 0.7080 | 0.6388 | 0.2321 | 0.3895 | 0.4871 | 0.4852 | 0.5625 | 0.6315 | 0.6823 | 0.5352 |
| | K8V2 | 0.6872 | 0.6204 | 0.2225 | 0.3856 | 0.4847 | 0.4928 | 0.5368 | 0.6058 | 0.6641 | 0.5222 |
| | K4V8 | 0.7125 | 0.6275 | 0.2326 | 0.3923 | 0.4761 | 0.4814 | 0.5580 | 0.6096 | 0.6550 | 0.5272 |
| | KV4 | 0.7013 | 0.6249 | 0.2322 | 0.4048 | 0.4627 | 0.4761 | 0.5474 | 0.6240 | 0.6368 | 0.5234 |
| | K4V2 | 0.6709 | 0.6038 | 0.2216 | 0.3885 | 0.4700 | 0.4519 | 0.5284 | 0.5732 | 0.6171 | 0.5028 |
| | K2V4 | 0.3566 | 0.3626 | 0.1986 | 0.2995 | 0.4186 | 0.0197 | 0.0099 | 0.0099 | 0.0091 | 0.1872 |
| | KV2 | 0.3507 | 0.3203 | 0.1983 | 0.2727 | 0.4308 | 0.0136 | 0.0144 | 0.0144 | 0.0136 | 0.1810 |
| Per-token-asym | KV8 | 0.7043 | 0.6379 | 0.2248 | 0.3866 | 0.4798 | 0.4913 | 0.5823 | 0.6331 | 0.6740 | 0.5349 |
| | K8V4 | 0.6969 | 0.6364 | 0.2402 | 0.3837 | 0.4676 | 0.4784 | 0.5671 | 0.6209 | 0.6717 | 0.5292 |
| | K8V2 | 0.4926 | 0.4979 | 0.0100 | 0.3732 | 0.4749 | 0.3616 | 0.3798 | 0.4200 | 0.4640 | 0.3860 |
| | K4V8 | 0.2489 | 0.2306 | 0.0000 | 0.2258 | 0.1591 | 0.0008 | 0 | 0 | 0.0008 | 0.0962 |
| | KV4 | 0.2377 | 0.2325 | 0.0000 | 0.2220 | 0.1469 | 0 | 0 | 0.0015 | 0.0015 | 0.0936 |
| | K4V2 | 0.2600 | 0.2323 | 0.0000 | 0.2258 | 0.0979 | 0.0038 | 0 | 0 | 0 | 0.0911 |
| | K2V4 | 0.2318 | 0.2335 | 0.0001 | 0.2201 | 0.1677 | 0.0023 | 0.0083 | 0.0045 | 0.0099 | 0.0976 |
| | KV2 | 0.2489 | 0.2372 | 0.0000 | 0.2249 | 0.1310 | 0.0023 | 0.0053 | 0.0106 | 0.0061 | 0.0963 |

| Quant. method | Precision | CEVAL | MMLU | TriviaQA | RACE | TruthfulQA | GSM8K 0-shot | 4-shot | 8-shot | 16-shot | Average |
|---|---|---|---|---|---|---|---|---|---|---|---|
| **Qwen2.5-7B-Instruct** | | | | | | | | | | | |
| BF16 | BF16 | 0.7949 | 0.7178 | 0.3239 | 0.4612 | 0.5104 | 0.7233 | 0.8059 | 0.8287 | 0.8218 | 0.6653 |
| KIVI | KV8 | 0.7949 | 0.7174 | 0.3235 | 0.4603 | 0.5092 | 0.721 | 0.7915 | 0.8249 | 0.8302 | 0.6637 |
| | K8V4 | 0.7979 | 0.7174 | 0.3222 | 0.4651 | 0.5104 | 0.7119 | 0.7915 | 0.8180 | 0.8226 | 0.6619 |
| | K8V2 | 0.7734 | 0.7035 | 0.3165 | 0.4459 | 0.4994 | 0.6581 | 0.7832 | 0.8059 | 0.8105 | 0.6440 |
| | K4V8 | 0.5780 | 0.5024 | 0.2757 | 0.3311 | 0.3660 | 0.0136 | 0.0076 | 0.0038 | 0.003 | 0.2312 |
| | KV4 | 0.5802 | 0.5028 | 0.2761 | 0.3206 | 0.3758 | 0.0182 | 0.0068 | 0.0038 | 0.003 | 0.2319 |
| | K4V2 | 0.5245 | 0.4704 | 0.2754 | 0.3167 | 0.3745 | 0.0152 | 0.0099 | 0.0053 | 0.0038 | 0.2217 |
| | K2V4 | 0.2719 | 0.2645 | 0.2742 | 0.2507 | 0.2399 | 0.0053 | 0.0015 | 0.0008 | 0.0008 | 0.1455 |
| | KV2 | 0.2756 | 0.2568 | 0.2741 | 0.2632 | 0.2338 | 0.0099 | 0.0038 | 0.0023 | 0 | 0.1466 |
| Per-token-asym | KV8 | 0.7883 | 0.7119 | 0.3192 | 0.4593 | 0.4957 | 0.7149 | 0.8044 | 0.8052 | 0.8203 | 0.6577 |
| | K8V4 | 0.7920 | 0.7117 | 0.2978 | 0.4545 | 0.5018 | 0.7111 | 0.7847 | 0.8044 | 0.8067 | 0.6516 |
| | K8V2 | 0.7169 | 0.6757 | 0.1127 | 0.4488 | 0.4957 | 0.577 | 0.7233 | 0.7453 | 0.7513 | 0.5830 |
| | K4V8 | 0.2192 | 0.2305 | 0.0000 | 0.2220 | 0.0318 | 0 | 0 | 0 | 0 | 0.0782 |
| | KV4 | 0.2400 | 0.2327 | 0.0000 | 0.2115 | 0.0171 | 0.0008 | 0.0015 | 0 | 0 | 0.0782 |
| | K4V2 | 0.2400 | 0.2301 | 0.0001 | 0.2172 | 0.0245 | 0.0023 | 0.0008 | 0 | 0 | 0.0794 |
| | K2V4 | 0.2273 | 0.2347 | 0.0001 | 0.2077 | 0.0575 | 0.0061 | 0.0068 | 0.0015 | 0.0015 | 0.0826 |
| | KV2 | 0.2489 | 0.2376 | 0.0000 | 0.2230 | 0.1346 | 0.0045 | 0.003 | 0.0076 | 0.0015 | 0.0956 |

| Quant. method | Precision | CEVAL | MMLU | TriviaQA | RACE | TruthfulQA | GSM8K 0-shot | 4-shot | 8-shot | 16-shot | Average |
|---|---|---|---|---|---|---|---|---|---|---|---|
| **Qwen2.5-Math-7B-Instruct** | | | | | | | | | | | |
| BF16 | BF16 | 0.4881 | 0.5383 | 0.0074 | 0.3464 | 0.4015 | 0.4109 | 0.8863 | 0.8870 | 0.8840 | 0.5389 |
| KIVI | KV8 | 0.4844 | 0.5379 | 0.0072 | 0.3397 | 0.3966 | 0.4041 | 0.8878 | 0.8878 | 0.8772 | 0.5359 |
| | K8V4 | 0.4874 | 0.5361 | 0.0071 | 0.3445 | 0.4002 | 0.4102 | 0.8886 | 0.8870 | 0.8840 | 0.5383 |
| | K8V2 | 0.4606 | 0.5291 | 0.0071 | 0.3426 | 0.4162 | 0.4139 | 0.8779 | 0.8802 | 0.8696 | 0.5330 |
| | K4V8 | 0.4428 | 0.5061 | 0.0073 | 0.2660 | 0.4100 | 0.0834 | 0.1501 | 0.2024 | 0.1259 | 0.2438 |
| | KV4 | 0.4368 | 0.5070 | 0.0074 | 0.2718 | 0.4284 | 0.0879 | 0.1516 | 0.1895 | 0.1236 | 0.2449 |
| | K4V2 | 0.4294 | 0.4862 | 0.0069 | 0.2699 | 0.4100 | 0.0819 | 0.1145 | 0.1433 | 0.1024 | 0.2272 |
| | K2V4 | 0.2712 | 0.2780 | 0.0059 | 0.2230 | 0.3941 | 0.0152 | 0.0061 | 0.0023 | 0.0008 | 0.1330 |
| | KV2 | 0.2741 | 0.2757 | 0.0057 | 0.2220 | 0.3501 | 0.0167 | 0.0023 | 0.003 | 0 | 0.1277 |
| Per-token-asym | KV8 | 0.3975 | 0.5905 | 0.6064 | 0.4038 | 0.4308 | 0.0728 | 0.3457 | 0.3685 | 0.3571 | 0.3970 |
| | K8V4 | 0.3878 | 0.5891 | 0.6035 | 0.4010 | 0.4443 | 0.0735 | 0.3450 | 0.3616 | 0.3632 | 0.3966 |
| | K8V2 | 0.3522 | 0.5590 | 0.5452 | 0.3971 | 0.3452 | 0.0462 | 0.3116 | 0.3397 | 0.3359 | 0.3591 |
| | K4V8 | 0.3804 | 0.5822 | 0.6016 | 0.4010 | 0.3831 | 0.0667 | 0.3252 | 0.351 | 0.3381 | 0.3810 |
| | KV4 | 0.3767 | 0.5803 | 0.5967 | 0.4038 | 0.3953 | 0.0622 | 0.3093 | 0.3146 | 0.3404 | 0.3755 |
| | K4V2 | 0.3470 | 0.5463 | 0.5372 | 0.3943 | 0.4211 | 0.0462 | 0.2631 | 0.2752 | 0.2911 | 0.3468 |
| | K2V4 | 0.2429 | 0.2401 | 0.0262 | 0.2900 | 0.2693 | 0.0121 | 0.0038 | 0.0045 | 0.0083 | 0.1219 |
| | KV2 | 0.2363 | 0.2351 | 0.0110 | 0.2766 | 0.1787 | 0.0121 | 0.0061 | 0.0091 | 0.0091 | 0.1082 |

Note: The first table block is under the heading **Qwen2.5-3B-Instruct-AWQ**.

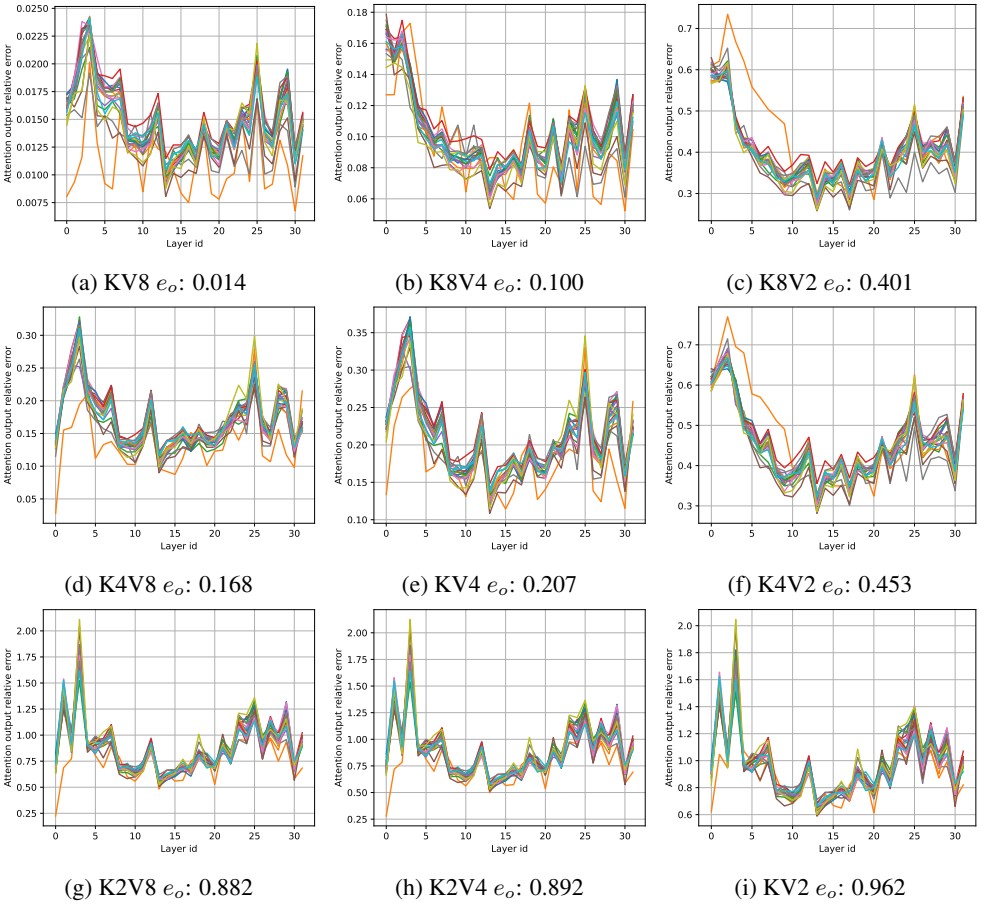

(a) KV8 $e_o$: 0.014      (b) K8V4 $e_o$: 0.100      (c) K8V2 $e_o$: 0.401

(d) K4V8 $e_o$: 0.168      (e) KV4 $e_o$: 0.207      (f) K4V2 $e_o$: 0.453

(g) K2V8 $e_o$: 0.882      (h) K2V4 $e_o$: 0.892      (i) KV2 $e_o$: 0.962

Figure 13: Layer-wise relative attention output error $e_o$ of **per-token-asym** KV cache quantization with simulated offline quantization and dequantization (without error accumulation) of the **Llama-3.1-8B-Instruct** model and the first 20 prompts in the **0-shot GSM8K** dataset. When the key quantization precision decreases to 2-bit, the layer-wise relative attention output error distribution significantly shifts. Especially, the errors of layer-3 and layer-1 are significantly larger than other layers.

## F. Layer-wise Attention Score and Relative Output Error

In this section, we visualize more layer-wise attention errors with KV cache quantization covering different LLMs, datasets, and KV cache quantization mode and precision. We select the first 20 prompts from the mathematical reasoning dataset GSM8K (Cobbe et al., 2021) and the AIGC multi-turn conversation dataset multiturn-softage (SoftAge-AI, 2024). Tested LLMs include Llama-3.1-8B-Instruct, Qwen2.5-7B-Instruct, and Mistral-7B-Instruct-v0.3. The layer-wise sensitivity to KV cache quantization of different LLMs are consistent to different prompts and datasets. Key cache quantization generally leads to the layer-wiser attention output error distribution shift. When the layer-wise attention error distribution significantly changes, the final model accuracy also dramatically degrades. For example, the perplexity and final generation accuracy of Qwen2.5-7B-Instruct dramatically degrades when the key quantization precision decreases to 4-bit and 2-bit with the KIVI or per-token-asym quantization mode as demonstrated in Table 2, 5, and 6. The attention distribution of it also significantly shifts as shown in Figure 16, 17, and 18. The most

As visualized in Figure 12, most layers of Qwen2.5-7B-Instruct have a high ratio of non-sparse retrieval heads, which are sensitive to low-precision key cache quantization as analyzed in Section 4.4. As a result, 4-bit or 2-bit key quantization leads to noticeable errors of attention score and critical KV identification in these layers with medium attention errors such as layer-1, 12, and 21.

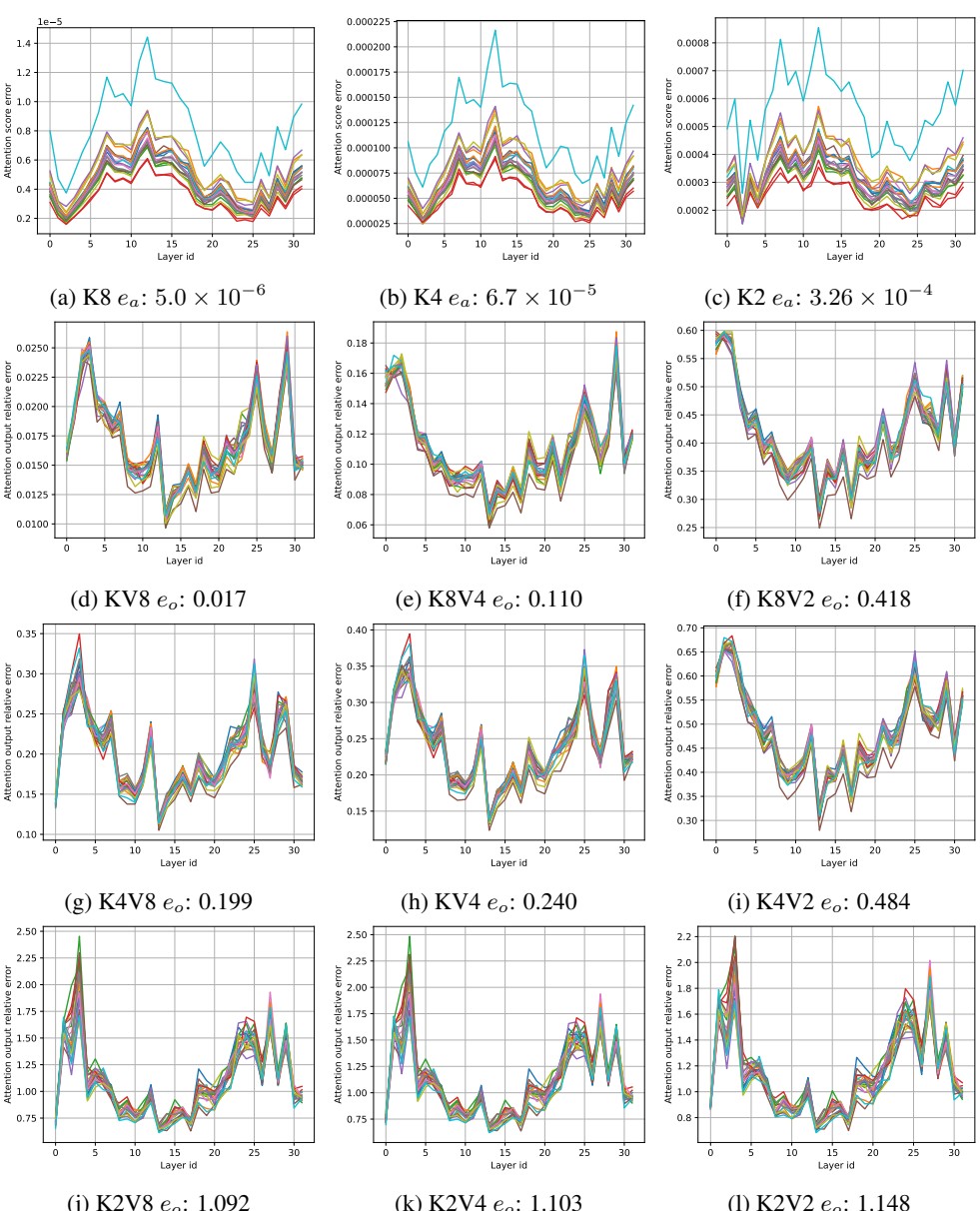

Figure 14: Layer-wise attention score errors $e_a$ and relative attention output error $e_o$ of **per-token-asym** KV cache quantization with simulated offline quantization and dequantization (without error accumulation) of the **Llama-3.1-8B-Instruct** model and the first 20 prompts in the **AIGC multiturn softage** dataset. When the key quantization precision decreases to 2-bit, the layer-wise relative attention output error distribution significantly shifts. Especially, the errors of layer-3, layer-1, and layer-27 are significantly larger than other layers.

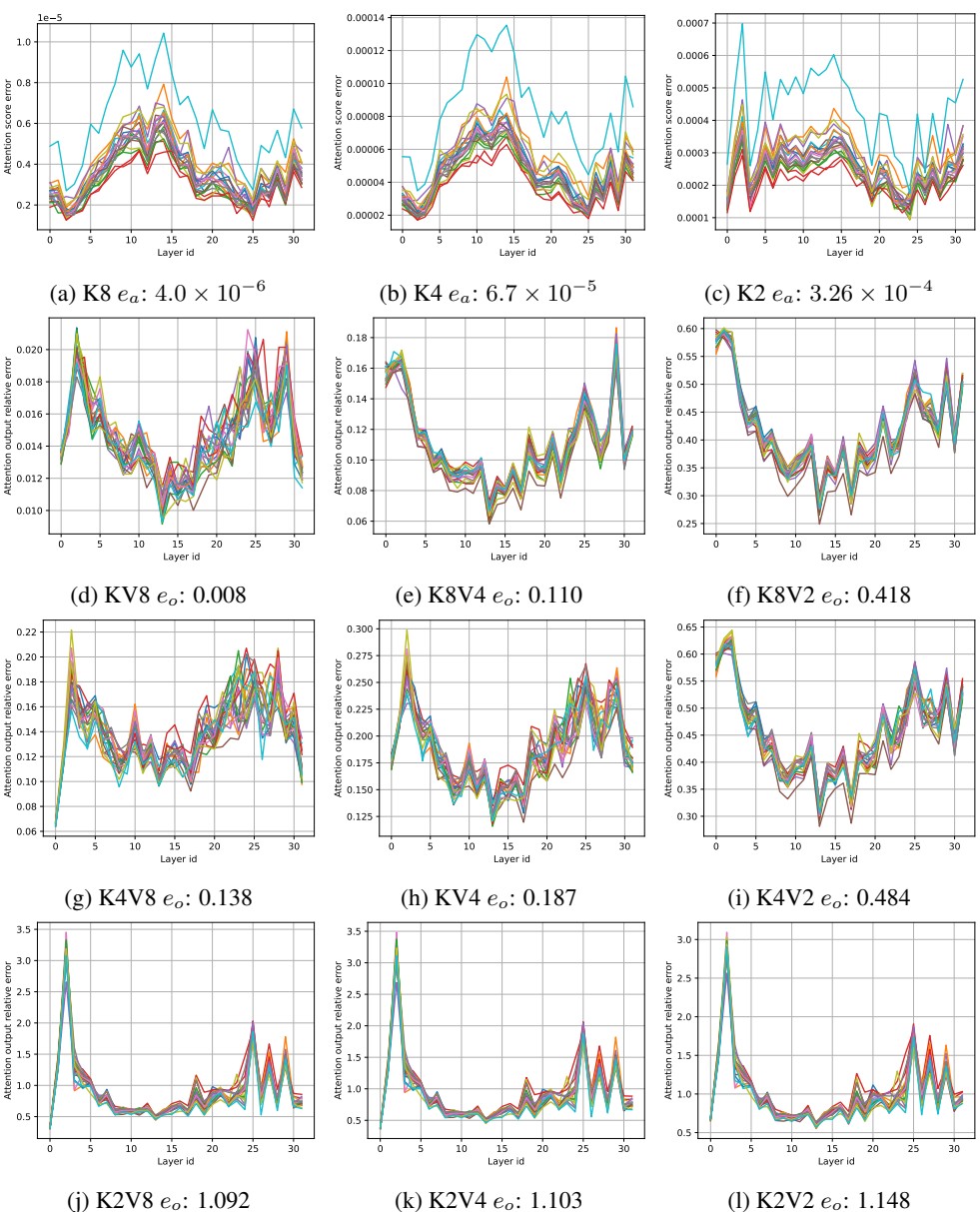

(a) K8 $e_a$: $4.0 \times 10^{-6}$  (b) K4 $e_a$: $6.7 \times 10^{-5}$  (c) K2 $e_a$: $3.26 \times 10^{-4}$

(d) KV8 $e_o$: 0.008  (e) K8V4 $e_o$: 0.110  (f) K8V2 $e_o$: 0.418

(g) K4V8 $e_o$: 0.138  (h) KV4 $e_o$: 0.187  (i) K4V2 $e_o$: 0.484

(j) K2V8 $e_o$: 1.092  (k) K2V4 $e_o$: 1.103  (l) K2V2 $e_o$: 1.148

Figure 15: Layer-wise attention score errors $e_a$ and relative attention output error $e_o$ of **key per-channel-asym and value per-token-asym** quantization with simulated offline quantization and dequantization (without error accumulation) of the **Llama-3.1-8B-Instruct** model and the first 20 prompts in the **AIGC multiturn softage** dataset. When the key quantization precision decreases to 2-bit, the layer-wise relative attention output error distribution significantly shifts. Especially, the errors of layer-2 and 27 are significantly larger than other layers.

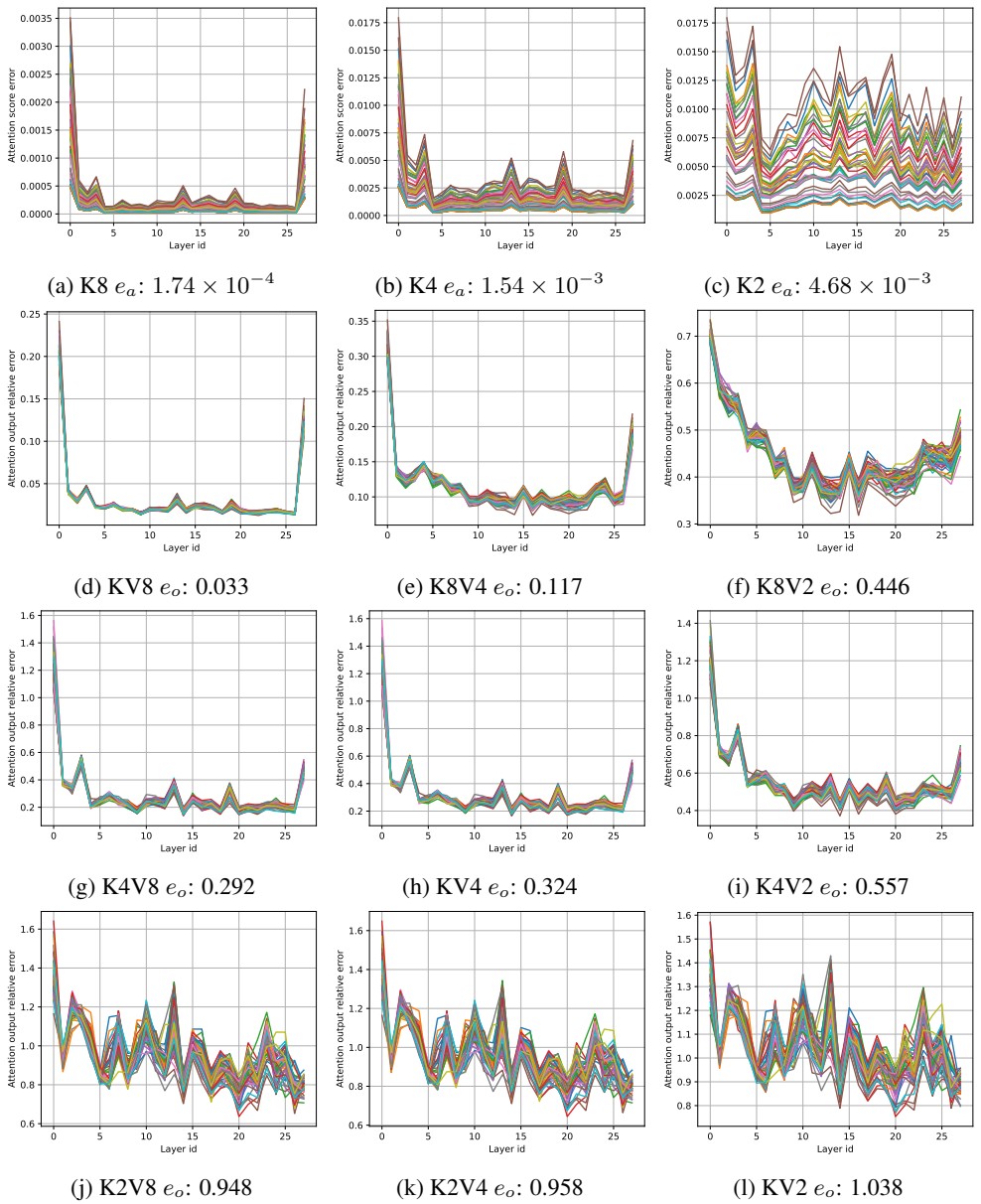

(a) K8 $e_a$: $1.74 \times 10^{-4}$  (b) K4 $e_a$: $1.54 \times 10^{-3}$  (c) K2 $e_a$: $4.68 \times 10^{-3}$

(d) KV8 $e_o$: 0.033  (e) K8V4 $e_o$: 0.117  (f) K8V2 $e_o$: 0.446

(g) K4V8 $e_o$: 0.292  (h) KV4 $e_o$: 0.324  (i) K4V2 $e_o$: 0.557

(j) K2V8 $e_o$: 0.948  (k) K2V4 $e_o$: 0.958  (l) KV2 $e_o$: 1.038

Figure 16: Layer-wise attention score $e_a$ and relative attention output error $e_o$ of **per-token-asym** KV cache quantization with simulated offline quantization and dequantization (without error accumulation) of the **Qwen2.5-7B-Instruct** model and the first 20 prompts in the **0-shot GSM8K** dataset. When the key quantization precision decreases to 4-bit or 2-bit, the layer-wise relative attention output error distribution significantly shifts. It also explains the performance degradation of Qwen2.5-7B-Instruct in the wikitext and other datasets. Especially, the errors of layer-3 and 13 are significantly larger than other layers. Note that in the 8-bit key cache quantization precision, only the first layer-0 and last layer-27 show significantly high errors, while in the 4-bit and 2-bit key cache quantization precision, the attention output errors of layer-3, 7, 10, 13, and 23 become noticeable compared with the first and last layers. Although these layers have relative simpler attention patterns as demonstrated in Figure 12, the low-precision 4-bit and 2-bit key cache quantization results in significantly token-level attention distribution shift.

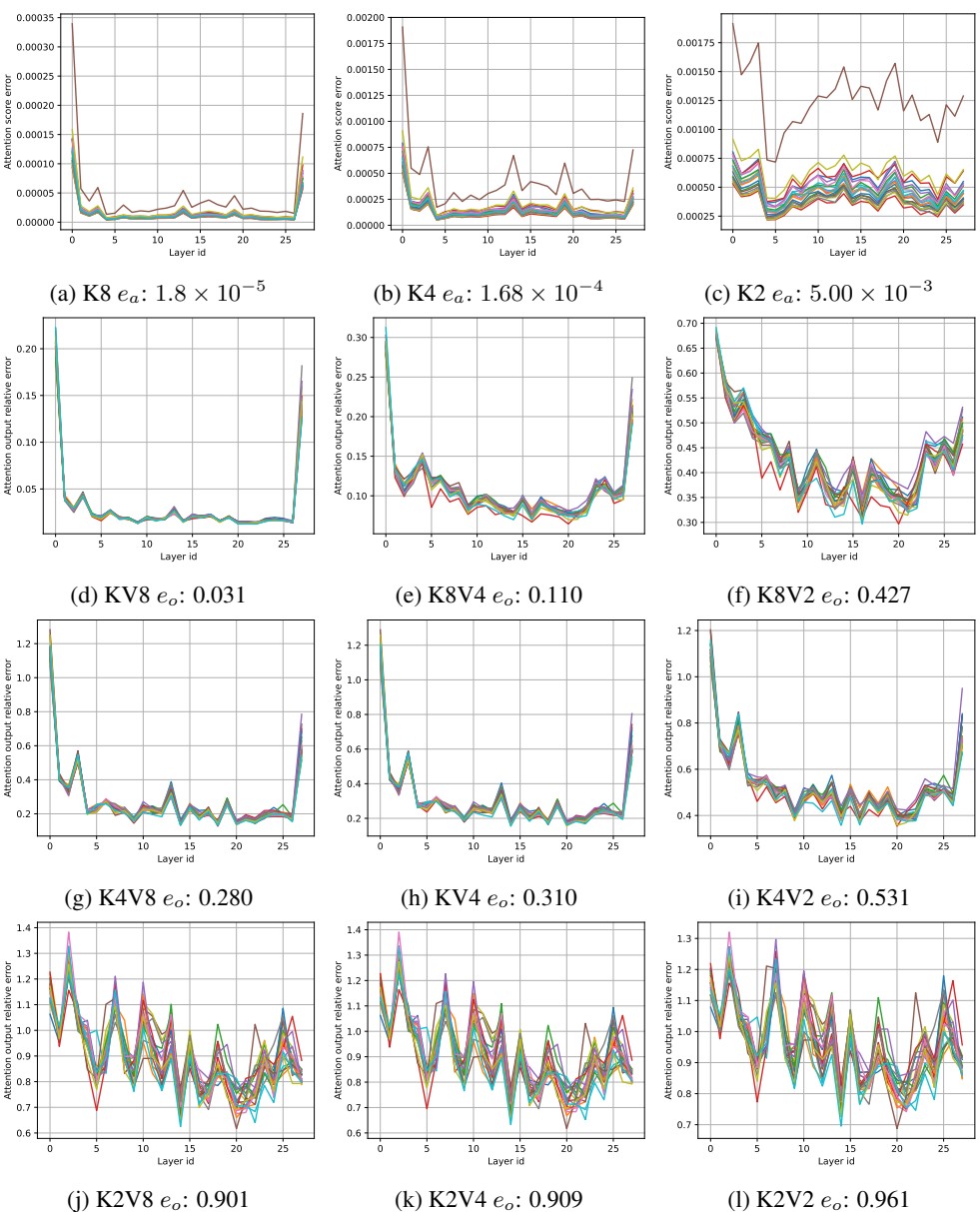

(a) K8 $e_a$: $1.8 \times 10^{-5}$     (b) K4 $e_a$: $1.68 \times 10^{-4}$     (c) K2 $e_a$: $5.00 \times 10^{-3}$

(d) KV8 $e_o$: 0.031     (e) K8V4 $e_o$: 0.110     (f) K8V2 $e_o$: 0.427

(g) K4V8 $e_o$: 0.280     (h) KV4 $e_o$: 0.310     (i) K4V2 $e_o$: 0.531

(j) K2V8 $e_o$: 0.901     (k) K2V4 $e_o$: 0.909     (l) K2V2 $e_o$: 0.961

Figure 17: Layer-wise attention score $e_a$ and relative attention output error $e_o$ of **per-token-asym** KV cache quantization with simulated offline quantization and dequantization (without error accumulation) of the **Qwen2.5-7B-Instruct** model and the first 20 prompts in the **AIGC multiturn softage** dataset. The layer-wise attention error shift is similar to Figure 16, indicating that the layer-wise sensitivity to KV cache quantization is independent of the input prompts and even domains.

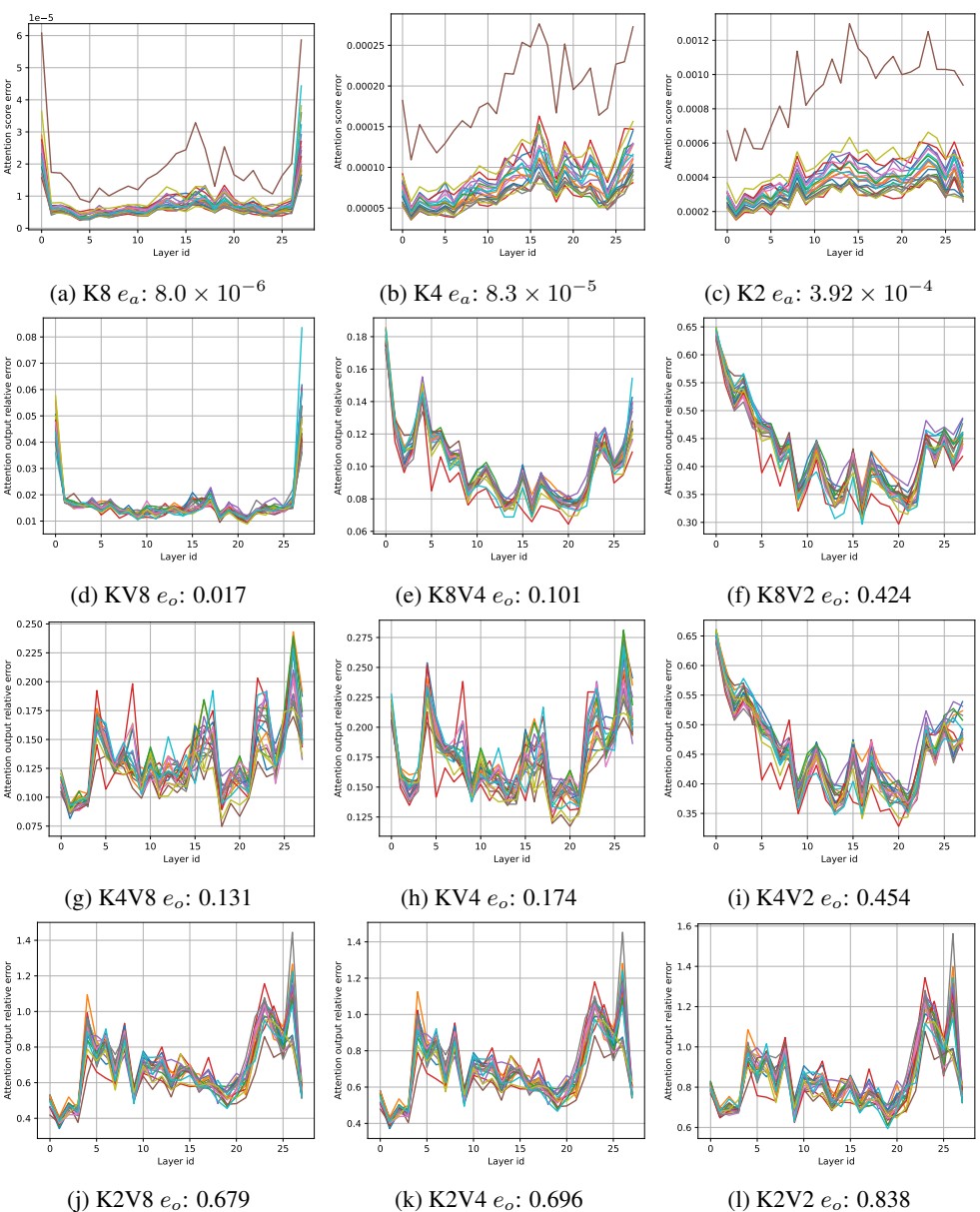

(a) K8 $e_a$: $8.0 \times 10^{-6}$
(b) K4 $e_a$: $8.3 \times 10^{-5}$
(c) K2 $e_a$: $3.92 \times 10^{-4}$

(d) KV8 $e_o$: 0.017
(e) K8V4 $e_o$: 0.101
(f) K8V2 $e_o$: 0.424

(g) K4V8 $e_o$: 0.131
(h) KV4 $e_o$: 0.174
(i) K4V2 $e_o$: 0.454

(j) K2V8 $e_o$: 0.679
(k) K2V4 $e_o$: 0.696
(l) K2V2 $e_o$: 0.838

Figure 18: Layer-wise attention score $e_a$ and relative attention output error $e_o$ of **key per-channel-asym and value per-token-asym** quantization with simulated offline quantization and dequantization (without error accumulation) of the **Qwen2.5-7B-Instruct** model and the first 20 prompts in the **AIGC multiturn softage** dataset. Key quantization along the channel dimension significantly affects the distribution of critical layers for 4-bit and 2-bit precision compared with those in Figure 17. The averaged attention output errors $e_o$ under the same KV precision pairs also dramatically reduced.

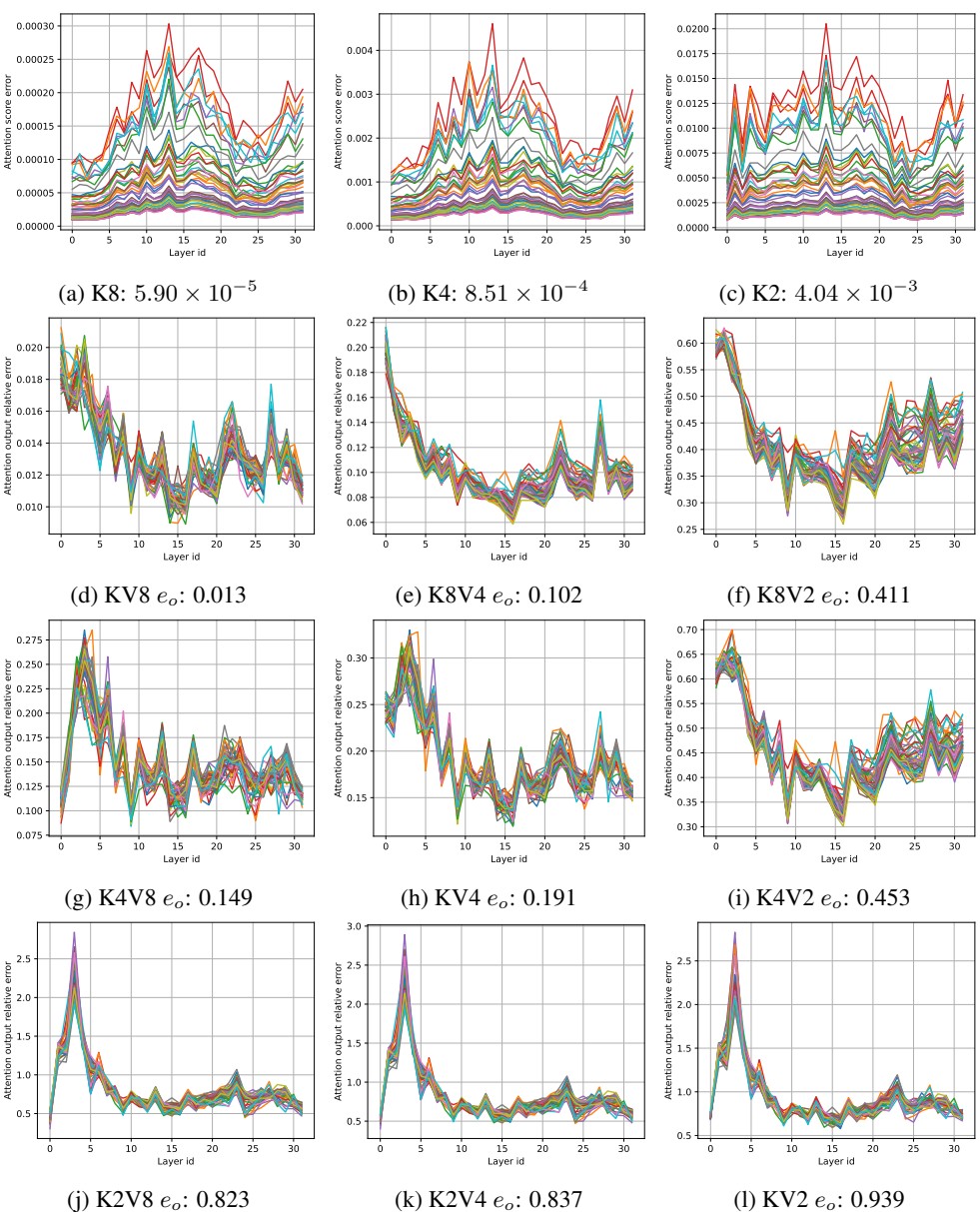

Figure 19: Layer-wise attention score $e_a$ and relative attention output error $e_o$ of **per-token-asym** KV cache quantization with simulated offline quantization and dequantization (without error accumulation) of the **Mistral-7B-Instruct-v0.3** model and the first 20 prompts in the **0-shot GSM8K** dataset. When the key quantization precision decreases to 2-bit, the layer-wise relative attention output error distribution significantly shifts. Especially, the errors of layer-1, 2, 3, and 4 are significantly larger than other layers.

Table 14: **KIVI-HQQ** KV cache quantization results of different precision and LLM models on the GSM8K few-shot CoTs as multiturn conversation dataset.

| Precision | GSM8K | | | Average | GSM8K | | | Average |
|---|---|---|---|---|---|---|---|---|
| | 4-shot | 8-shot | 16-shot | | 4-shot | 8-shot | 16-shot | |
| | **Llama-3-8B-Instruct** | | | | **Qwen2.5-7B-Instruct** | | | |
| BF16 | 0.7794 | 0.8006 | 0.7847 | 0.7882 | 0.6998 | 0.7377 | 0.7506 | 0.7294 |
| KV8 | 0.7801 | 0.8006 | 0.7824 | 0.7877 | 0.6801 | 0.7369 | 0.7460 | 0.7210 |
| K8V4 | 0.7688 | 0.7862 | 0.7809 | 0.7786 | 0.6793 | 0.724 | 0.7468 | 0.7167 |
| K8V2 | 0.7566 | 0.7763 | 0.7642 | 0.7657 | 0.6801 | 0.7491 | 0.7437 | 0.7243 |
| K4V8 | 0.7445 | 0.7695 | 0.7498 | 0.7546 | 0.0076 | 0.0038 | 0.0053 | 0.0056 |
| KV4 | 0.7422 | 0.7688 | 0.7384 | 0.7498 | 0.0038 | 0.0053 | 0.0023 | 0.0038 |
| K4V2 | 0.7346 | 0.7437 | 0.7430 | 0.7404 | 0.0061 | 0.0023 | 0.0038 | 0.0041 |
| K2V4 | 0.0152 | 0.0167 | 0.0159 | 0.0159 | 0.0045 | 0.0045 | 0.0023 | 0.0038 |
| KV2 | 0.0159 | 0.0144 | 0.0152 | 0.0152 | 0.0023 | 0.0015 | 0.003 | 0.0023 |
| | **Mistral-7B-Instruct-v0.3** | | | | **Qwen2.5-Math-7B-Instruct** | | | |
| BF16 | 0.5019 | 0.4890 | 0.4973 | 0.4961 | 0.8901 | 0.8666 | 0.8658 | 0.8742 |
| KV8 | 0.5042 | 0.4890 | 0.4966 | 0.4966 | 0.8901 | 0.8658 | 0.8666 | 0.8742 |
| K8V4 | 0.5064 | 0.4890 | 0.4913 | 0.4956 | 0.8931 | 0.8688 | 0.8628 | 0.8749 |
| K8V2 | 0.4837 | 0.4663 | 0.4632 | 0.4711 | 0.8719 | 0.8741 | 0.8491 | 0.8650 |
| K4V8 | 0.4754 | 0.4701 | 0.4534 | 0.4663 | 0.0500 | 0.0576 | 0.0697 | 0.0591 |
| KV4 | 0.4875 | 0.4754 | 0.4822 | 0.4817 | 0.0455 | 0.0516 | 0.0796 | 0.0589 |
| K4V2 | 0.4428 | 0.4503 | 0.4579 | 0.4503 | 0.0425 | 0.0516 | 0.0607 | 0.0516 |
| K2V4 | 0.0258 | 0.0288 | 0.0250 | 0.0265 | 0.0023 | 0 | 0 | 0.0008 |
| KV2 | 0.0190 | 0.0220 | 0.0208 | 0.0206 | 0.0023 | 0.0008 | 0.0015 | 0.0015 |
| | **Qwen2.5-3B-Instruct** | | | | **Qwen2.5-14B-Instruct** | | | |
| BF16 | 0.5732 | 0.5997 | 0.6459 | 0.6063 | 0.7536 | 0.7862 | 0.8180 | 0.7859 |
| KV8 | 0.583 | 0.6035 | 0.6353 | 0.6073 | 0.7491 | 0.7877 | 0.8158 | 0.7842 |
| K8V4 | 0.5603 | 0.5967 | 0.6513 | 0.6028 | 0.7551 | 0.7953 | 0.8264 | 0.7923 |
| K8V2 | 0.5133 | 0.5481 | 0.5997 | 0.5537 | 0.743 | 0.7733 | 0.8029 | 0.7731 |
| K4V8 | 0.5118 | 0.5057 | 0.5049 | 0.5075 | 0.7430 | 0.7779 | 0.7998 | 0.7736 |
| KV4 | 0.5080 | 0.4845 | 0.4837 | 0.4921 | 0.7339 | 0.7908 | 0.8112 | 0.7786 |
| K4V2 | 0.4587 | 0.4124 | 0.4170 | 0.4294 | 0.7475 | 0.7733 | 0.7953 | 0.7720 |
| K2V4 | 0.0083 | 0.0061 | 0.0136 | 0.0093 | 0.0220 | 0.0144 | 0.0174 | 0.0179 |
| KV2 | 0.0061 | 0.0076 | 0.0076 | 0.0071 | 0.0288 | 0.0152 | 0.0167 | 0.0202 |
| | **Qwen2.5-3B-Instruct-AWQ** | | | | **Qwen2.5-32B-Instruct** | | | |
| BF16 | 0.5656 | 0.6209 | 0.6399 | 0.6088 | 0.7619 | 0.7809 | 0.7961 | 0.7796 |
| KV8 | 0.5686 | 0.6149 | 0.6550 | 0.6128 | 0.7650 | 0.7877 | 0.8021 | 0.7849 |
| K8V4 | 0.5747 | 0.608 | 0.6406 | 0.6078 | 0.7726 | 0.7801 | 0.7998 | 0.7842 |
| K8V2 | 0.5466 | 0.5694 | 0.6149 | 0.5770 | 0.7384 | 0.7703 | 0.7877 | 0.7655 |
| K4V8 | 0.4845 | 0.4564 | 0.4443 | 0.4617 | 0.7597 | 0.7794 | 0.8135 | 0.7842 |
| KV4 | 0.4845 | 0.4807 | 0.4352 | 0.4668 | 0.7680 | 0.7718 | 0.8097 | 0.7832 |
| K4V2 | 0.4177 | 0.3730 | 0.3518 | 0.3808 | 0.7559 | 0.7733 | 0.7801 | 0.7698 |
| K2V4 | 0.0114 | 0.0091 | 0.0053 | 0.0086 | 0.0379 | 0.0281 | 0.0311 | 0.0324 |
| KV2 | 0.0167 | 0.0114 | 0.0129 | 0.0137 | 0.0258 | 0.0136 | 0.0311 | 0.0235 |

