# OpenReview forum: "KVTuner: Sensitivity-Aware Layer-Wise Mixed-Precision KV Cache Quantization for Efficient and Nearly Lossless LLM Inference"
_ICML.cc/2025/Conference — ICML 2025 poster_

### Official Review · Reviewer_uJyY · 2025-03-12

**Overall Recommendation:** 3

**Summary:**

This paper proposes a multi-objective optimization-based algorithm to search for the optimal layer-wise mixed precision KV cache quantization configuration. The authors observe that key caches generally require more bits for quantization than value caches, and thus propose allocating more bits to the key cache. Additionally, they find that different layers exhibit varying sensitivity to KV cache quantization. To address this, they apply mixed precision across different layers and employ a search algorithm to identify the optimal configuration. To reduce the search space, they incorporate pruning and clustering strategies. Experimental results across various LLMs and tasks demonstrate the accuracy and latency benefits of their approach.

## update after rebuttal

The authors addressed my concerns regarding the baseline comparison and provided additional experimental results to support the effectiveness of their method. Therefore, I will raise my rating to a weak accept.

**Claims And Evidence:**

The authors make three claims:

1. The key cache is more important than the value cache.

2. Different layers exhibit varying sensitivity to KV cache quantization.

3. Their proposed method outperforms baseline methods that use static quantization bits for all KV caches, such as KIVI.

However, claims 1 and 3 are problematic.

For claim 1, Table 4 shows that in some layers, the value cache is actually more important than the key cache. The authors should explain why this occurs and clarify whether claim 1 holds universally or if it depends on specific conditions.

For claim 3, it is not entirely clear that their method outperforms KIVI. In Table 5, KVTuner-C4.90 performs slightly worse than KIVI-4 for Llama-3.1-8B-Instruct, and KVTuner-C3.44 is worse than KIVI-4 for Qwen2.5-3B-Instruct. Moreover, the authors do not include a baseline of KIVI-3, making it difficult to provide a fair comparison between KIVI and the proposed method. For instance, although KVTuner-C3.25 outperforms KIVI-2 for Llama-3.1-8B-Instruct, KVTuner-C3.25 uses a higher bitwidth than KIVI-2, making the comparison unfair. This issue also applies to Table 6.

Regarding latency, the baseline is unclear. Is KV8 referring to KIVI-8 or just standard 8-bit quantization? KIVI-n is used in Tables 5 and 6 but not in Table 7, which adds to the confusion. If KV8 is not KIVI-8, why not compare with KIVI-8? Furthermore, Llama2-7B is used for the latency comparison, but it is not included in the accuracy comparison, leaving the accuracy difference between KV8 and KVTuner-C6 unknown. Without this information, the latency comparison lacks context. It is difficult to assess whether the proposed method provides a better latency-accuracy trade-off than the baselines.

**Essential References Not Discussed:**

All essential references are discussed.

**Experimental Designs Or Analyses:**

The accuracy experiments have the issue of lacking appropriate baselines with roughly the same quantization bitwidth as their proposed method.

The latency experiment in Section 6.3 is problematic, as it does not provide the accuracy results for Llama2-7B, making it difficult to assess the accuracy-latency trade-off.

In Section 6.5, it is unclear which points in Figure 18 correspond to the unified precision configuration.

**Methods And Evaluation Criteria:**

The proposed method is reasonable, given the varying importance of key/value caches and the different layer-wise sensitivity to KV cache. However, the evaluation lacks clarity, as the accuracy benefits are subtle, and the latency comparison, as previously mentioned, is problematic. Additionally, since KV cache quantization is particularly important for long-context generation, the authors should evaluate their method on long-context benchmarks, such as LongBench, to more effectively validate its performance.

**Other Comments Or Suggestions:**

Table 4 typo: Pateto -> Pareto

**Other Strengths And Weaknesses:**

1. Most experiments in this paper mention the use of per-token KV cache quantization, such as in Figures 2, 5, and 6. However, as demonstrated in KIVI, per-token key quantization performs significantly worse than per-channel quantization. Why not consistently apply per-channel quantization to the key?

2. Table 2 compares word-perplexity for the KIVI-HQQ implementation. However, in the HQQ implementation, both the key and value are quantized per-token. Since using per-token quantization for the key naturally leads to greater loss compared to per-channel quantization, it is unreliable to draw conclusions about whether the key is more important than the value when per-channel quantization is not applied to the key and per-token quantization is used for the value.

3. Section 4.5 aims to conclude that layer-wise sensitivity to KV cache quantization is an inherent characteristic of LLMs. However, the analysis is based only on prompts for math problems, and it is unclear whether this finding applies to general prompts, such as non-math tasks like retrieval and summarization.

4. Although the search space is greatly reduced, it is still considerably large (15,625, as mentioned in Line 319). The search cost may still be high, yet there is no explanation provided regarding the search cost, such as the total search time.

**Questions For Authors:**

1. What is the search cost of the proposed method?

2. The accuracy comparison is unreliable, as it lacks appropriate baselines with competitive quantization bitwidths for a fair comparison. Additionally, the accuracy improvement appears to be subtle.

3. The latency comparison is unreliable, as it lacks the accuracy results for the Llama2-7B model, making it impossible to assess the latency-accuracy trade-off.

**Relation To Broader Scientific Literature:**

The attempt to apply a multi-objective optimization algorithm to KV cache quantization in this work has the potential to broaden the application of MOO.

**Theoretical Claims:**

All proofs are correct.

---

> ### Author Rebuttal · Authors · 2025-04-01
>
> We sincerely thank you for your thorough feedback. Below, we address the concerns raised and outline revisions to improve the clarity and rigor of our work.
>
> ---
> # 1. Key Cache Importance
> * The reviewer correctly observes that in certain layers (e.g. Layer 0,1,2,31 of Llama-3.1-8B-Instruct), the proposed intra-layer KV cache precision pair pruning algorithm selects K4V8 rather than K8V4. We will clarify in the revised manuscript that Claim 1 ("key cache is generally more important") reflects an overall trend across layers. KVTuner assigns more bits to value cache than key cache in specific layers.
> Those layers where higher value bitwidths outperform higher key bitwidths normally have more streaming head patterns than retrieval heads, where key is more robust to low-bit quantization. Therefore, more memory bitwidths should be assigned to value cache, which is more sensitive in this case.
>
> * **This phenomenon verifies our observation and theoretical analysis about the strong correlation of KV cache quantization errors and attention patterns in Section 4.4.** In addition, the proposed KVTuner is adaptive to the inherent model structure patterns and the proposed layer-wise KV cache quantization precision tuning makes sense.
>
> ---
> # 2. Comparison with KIVI
> We appreciate the reviewer’s attention to fairness in comparisons. **KVTuner offers more flexible accuracy and efficiency tradeoff, which is not available in the static, uniform and non mixed-precisionbaselines KIVI and per-token-asym.**
>
> * The accuracy of KVTuner-C3.44 only decreases by 0.52% but the memory usage is reduced by 14% compared with KIVI-4 in Qwen2.5-3B-Instruct. We also push the frontier of nearly lossless KV cache compression (only 0.04% accuracy loss than the BP16 baseline) to 3.44-bit in this model.
> * In Table 11 (Page 24~25), we compare more uniform precision including K8V4 (C6), K8V2 (C5), and K4V2 (C3). Llama-3.1-8B-Instruct and Mistral-7B-v0.3 are more robust to low bit key quantization than Qwen2.5-7B-Instruct. However, Qwen2.5-7B-Instruct outperforms others in terms of accuracy.
> * In Table 5 and 6 (Page 8), the baselines INT4 KIVI and per-token-asym quantization lead to significant (67% and 26%) accuracy degradation in Qwen2.5-7B-Instruct. However, **KVTuner successfully reduces the accuracy loss to 16% with lower 3.92-bit memory usage and 0.18% with similar 4-bit memory usage, respectively. It indicates that the accuracy improvement of KVTuner is noticeable and KVTuner offers more robustness and flexibility to sensitive but powerful models.**
> * Figure 5 also visualizes the accuracy and equivalent KV cache bitwidth of different layer-wise KV cache precision pairs during KVTuner offline searching. The red circles are uniform precision across all layers. From Figure 5, we can easily observe more Pareto-optimal layer-wise KV cache precision pairs than uniform ones. **Especially, the accuracy of the uniform KV4, K4V2, and K2V4 is around 0% in Qwen2.5-7B-Instruct, while the searched equivalent 4-bit and 3-bit configs of KVTuner achieve 80% and 50% accuracy, respectively. It is a huge improvement in terms of accuracy with similar memory usage.**
> * Latency Baseline Clarification
> In Table 7, "KV8" refers to 8-bit KIVI quantization. We thus report the total model-level throughput comparison of Llama-3.1-8B-Instruct using the searched config in Table 5.  The hardwares are Nvidia RTX 4090 24G. Compared with KIVI-KV8, the throuhgput of KVTuner-C3.25 can be improved by 16.79%~21.25%.
> |BS, inputLen|KV8(baseline)|K8V4|KV4|K4V2|KVTuner-C4.92|KVTuner-C3.25|
> |-|-|-|-|-|-|-|
> |64,128|3836|4193|4567|4697|4240(10.53%)|4652(**21.25%**)|
> |8,1024|549|597|632|645|600(9.22%)|641(**16.79%**)|
> ---
> # 3. Long context evaluation
> We compare KVTuner with the basedlines in the 20 Longbench datasets and the averaged scores are as below. The conclusion is that KVTuner push the nearly lossless long context generation to 3.25-bit.
>
> ### Qwen2.5-7B-Instruct KIVI
>  |BF16|KIVI8|KIVI-K8V4|KIVI4|KVTuner-C4.92|KVTuner-C3.25|
> |-|-|-|-|-|-|
> |0.7956|0.7992|0.8001|0.7723|0.7956|0.7903|
>
> ### Qwen2.5-7B-Instruct per-token-asym
>
> |BF16|KV8|K8V4|KV4|KVTuner-C5.0|KVTuner-C4.0|
> |-|-|-|-|-|-|
> |0.7956|0.7971|0.7953|0.6343|0.8005|0.7960|
>
> ---
> # 4. Per-channel vs. per-token quantization
> KIVI-HQQ also supports key per-channel quantization by tuning the axis_key config. KIVI requires new operators and careful management as discussed in Line 58~69. In contrast, per-token-asym can be easily implemented and is supported in common inference frameworks such as LMDeploy.
>
> ---
> # 5. Layer-wise sensitivity analysis
> We also study the layer-wise sensitivity to KV cache quantization of Llama-3.1-8B-Instruct and Qwen2.5-7B-Instruct with both KV per-token-asym and KIVI like quantization modes on the non-math AIGC multiturn softage dataset in Figure 9 (Page 17), 10 (Page 18), 12 (Page 20), and 13 (Page 21). The sensitive layers are consistent with those in math tasks e.g. Figure 8.

---

> > ### Comment · Reviewer_uJyY · 2025-04-04
> >
> > Thank you for your response—it addressed all of my concerns, and I will be raising my score.

---

> > > ### Author Response · Authors · 2025-04-05
> > >
> > > Dear Reviewer,
> > >
> > > Thank you sincerely for your thoughtful feedback and for recognizing our efforts to address the concerns raised in your initial review. We deeply appreciate your constructive critique, which has significantly strengthened the rigor and clarity of our work. Your insights, particularly on baseline comparisons, evaluation fairness, and long-context validation, have been invaluable in refining our methodology and presentation.
> > >
> > > We are committed to incorporating all promised revisions into the final manuscript, ensuring the paper meets the high standards of the publication. Your expertise and time invested in reviewing our work are greatly acknowledged and appreciated.
> > >
> > > Thank you once again for your support and for guiding us toward a stronger contribution to the field.
> > >
> > > Best regards,
> > >
> > > Authors of the paper 11535

---

### Official Review · Reviewer_FF8e · 2025-03-12

**Overall Recommendation:** 3

**Summary:**

This paper proposed an innovative quantization technique for KV caches, which can reduce the inference throughput with a negligible quality drop in the output.

This paper's key insight is that the key cache is more important than the value cache in terms of reducing the quantization error. Its key contribution is an adaptive framework called KVTuner that can tune the KV quantization configurations offline and use them online for different objectives.

The experiments show that the solution can produce similar quality compared with the state-of-the-art solutions. And its inference efficiency is higher than a default KV8 quantization.

## Update after rebuttal
Thank the authors for the clarifications. I will maintain the decision of "weak accept".

**Claims And Evidence:**

The key insight of the paper (the key cache is more important than the value cache in terms of reducing the quantization error) is backed up with good experimental results (20 samples from the standard dataset on an up-to-date llama 8b model).

**Essential References Not Discussed:**

N/A

**Experimental Designs Or Analyses:**

The experimental design of quality evaluation is comprehensive and good.

However, the design of the efficiency evaluation is not clear (because of the problems mentioned above)

**Methods And Evaluation Criteria:**

The evaluation section contains 2 parts: quality evaluation and efficiency evaluation.

The quality evaluation is comprehensive and the metrics are suitable for the use case.

However, the efficiency evaluation needs to be improved on the following points:

1. The setup of the benchmark is not clear. The paper does not mention how the baseline and their solution are implemented. Is there any new CUDA kernel in the paper's solution? Is the baseline using state-of-the-art attention frameworks like FlashAttention?
2. The definition of "throughput" is not clear. Does it refer to the speed of token generation (i.e., tokens per second), or number of finished requests per second, or something else?

**Other Comments Or Suggestions:**

N/A

**Other Strengths And Weaknesses:**

N/A

**Questions For Authors:**

In Table 7, the 8K FP16 setup has an OOM error. I'm wondering why it could happen. 8K context length only corresponds to ~1GB of KV cache when using llama-7B models. If the GPU is 48GB, how could it be OOM? Could it be because there is something wrong with the experiment?

**Relation To Broader Scientific Literature:**

There are plenty of works focusing on KV cache compressions. Some of them suggest the key cache is less important than the value cache [1], and some of them suggest we should have different quantization methods for the key cache and value cache respectively [2].

The key claim in this paper seems to conflict with those prior works. Please discuss the difference and the potential reason.


References
----
[1] Zhao, Yilong, et al. "Atom: Low-bit quantization for efficient and accurate llm serving, 2024." URL https://arxiv. org/abs/2310.19102.
[2] Liu, Zirui, et al. "Kivi: A tuning-free asymmetric 2bit quantization for kv cache." arXiv preprint arXiv:2402.02750 (2024).

**Theoretical Claims:**

The theoretical claims and the algorithm design make sense and there are no obvious problems.

---

> ### Author Rebuttal · Authors · 2025-04-01
>
> Thank you for your thorough review and constructive feedback. We sincerely appreciate your acknowledgment of the innovativeness and feasibility of the proposed methodology and theoretical analysis of attention patterns to KV cache quantization, as well as your recognition of the comprehensive implementation of the experimental investigations.  Below, we address your concerns and suggestions point by point:
>
> ---
> # 1. Efficiency evaluation clarifications
> We acknowledge the lack of clarity in our efficiency evaluation setup. Here are key clarifications:
> * Baseline Implementation: In Table 7, the static SOTA baseline KIVI and our solutions are both based on the official KIVI code with flashattention enabled in their kernels during decoding. KV8 and KV4 are KIVI-8 and KIVI-4, respectively. We will correct them in the revised version. We slightly modify the KIVI CUDA kernels to support INT8 and mixed precision of key and value cache. KIVI, per-token-asym, and our KVTuner is a online static method and compatible with FlashAttention. In addition, It uses lightweight post-training calibration and offline bit-width selection. This design choice ensures compatibility with existing inference frameworks. However, other online mixed precision KV cache quantization methods with attention scores based online token importance estimation is normally not compatible with flashattention.
>
> * New GQA efficiency results with KVTuner: We modify the KIVI code to support GQA models including Llama-3.1-8B-Instruct. We thus report the total model-level throughput comparison of Llama-3.1-8B-Instruct using the searched config in Table 5.  The hardwares are Nvidia RTX 4090 24G. **Compared with KIVI-KV8, the throuhgput of KVTuner-C3.25 can be improved by 16.79%~21.25%.**
> |BS, inputLen|KV8(baseline)|K8V4|KV4|K4V2|KVTuner-C4.92|KVTuner-C3.25|
> |-|-|-|-|-|-|-|
> |64,128|3836|4193|4567|4697|4240(10.53%)|4652(21.25%)|
> |16,512|1102|1205|1275|1304|1239(12.41%)|1296(17.55%)|
> |8,1024|549|597|632|645|600(9.22%)|641(16.79%)|
>
> * Throughput definition:  We follow the same settings and definitions of KIVI. Throughput is defined as the the number of tokens generated per second (measured end-to-end, including quantization/dequantization overhead). For example, if the batch size is 128 and one generation step takes 50ms, the throughput is 128 * 1000 / 50 = 2560 tokens/s.
>
> ---
> # 2. Discussion of conflicting literature
> We appreciate the opportunity to clarify our position relative to prior works (Atom and KIVI):
> Atom claims that KV cache is more amerable to quantization than activation matrices in Section 4.4 and utilizes INT4 precision for both key and value. **The importance difference of key and value is not clearly discussed in Atom. KIVI implicitly implies that key is more important than value by applying more complex per-channel per-group quantization to key and simple per-token quantization to value.**
>
> The conclusion that key cache is normally more important than key cache is validated with extensive empirical studies, which include final perplexity on the **wikitext** dataset in Table 2, layer-wise attention errors on the **GSM8K** dataset in Table 3 (Page 4), final model accuracy on the **general CEVAL, MMLU, TriviaQA, RACE, and TruthfulQA** datasets with both **per-token-asym and KIVI** quantization modes in Table 11 (Page 25), layer-wise attention score and output errors with the key per-channel-asym and value per-token-asym quantization on **AIGC** multiturn softage dataset of **Llama-3.1-8B-Instruct and Qwen2.5-7B-Instruct** in Figure 10 (Page 18) and Figure 13 (Page 21).
>
> ---
> # 3. OOM with 8K sequence
> We use 4 batch size with 8K tokens (~4GB). The OOM error for the 8K FP16 setup may arise from an unoptimized memory allocation strategy in our prototype implementation (e.g., redundant intermediate tensors were not freed). The default efficiency testing in the KIVI repo does not enable FlashAttention during prefilling, which may also result in the OOM issue with 8K long sequence.
>
> ---
> # Conclusion
> The proposed layer-wise KV cache precision pair tuning naturally suits the layer-wise sensitivity to KV cache quantization, making KVTuner a practical solution to reduce the memory usage and improve inference efficiency of LLMs with various sensitivities. KVTuner successfully pushes the nearly lossless KV cache quantization in complex mathematical and sciencitific tasks to 3.25-bit for Llama-3.1-8B-Instruct and 4-bit for sensitive Qwen2.5-7B-Instruct. KVTuner also greatly narrows the performance difference between the simple per-token-asym and accurate KIVI quantization modes, even when using overall similar low-precision settings.
>
> Many KV cache quantization approaches are proposed recently, but the correlation to LLM attention patterns is not well studied. We theoretically prove that the sparse streaming heads are robust to KV cache quantization than sensitive retrieval heads, which is the cause of layer-wise sensitivity to KV cache quantization.

---

### Official Review · Reviewer_fdwh · 2025-03-15

**Overall Recommendation:** 2

**Summary:**

The authors propose KVTuner, a sensitivity-aware layer-wise mixed-precision KV cache quantization framework for LLM inference. KVTuner addresses key challenges in KV cache quantization, including layer-wise sensitivity to quantization errors, high overhead of fine-grained online adjustments, and inflexibility across different LLM architectures. Instead of applying uniform quantization across all layers, KVTuner performs an offline search for optimal layer-wise key and value precision pairs (e.g., K8V4, K4V2) using multi-objective optimization (MOO). This search considers both memory constraints and model accuracy. The precomputed precision pairs are then applied directly during inference, reducing computational overhead while maintaining nearly lossless accuracy.

**Claims And Evidence:**

The paper claims that KVTuner significantly improves LLM inference efficiency while maintaining accuracy close to full-precision KV caching. As shown in Table 7, KVTuner-C6 achieves a 38.3% throughput improvement compared to KV8, and KVTuner-C3 achieves an even higher 76.4% improvement. However, the selection method may introduce additional computational complexity. In lines 275-295, the authors discuss how KVTuner avoids online decision-making overhead.

**Essential References Not Discussed:**

The paper does not discuss alternative mixed-precision approaches that incorporate token-importance ranking for KV selection.

**Experimental Designs Or Analyses:**

The experiments comprehensively evaluate KVTuner across different models (Llama-3.1-8B, Qwen2.5-7B, Mistral-7B) and various KV precision configurations. The results in Table 5 show that KVTuner maintains accuracy while achieving significant memory savings. However, a few aspects could be further explored.

**Methods And Evaluation Criteria:**

The methodology is well-structured and based on a layer-wise sensitivity analysis of KV cache quantization. The evaluation uses standard mathematical reasoning benchmarks such as GSM8K and GPQA, which are appropriate for testing the impact of quantization errors. However, as noted in lines 220-250, additional profiling of the computational overhead per layer and the impact on inference latency in real-world applications (e.g., batched inference on vLLM) would strengthen the claims. A comparison of layer-wise FLOP costs before and after applying KVTuner’s selection would provide a clearer picture of its computational efficiency.

**Other Comments Or Suggestions:**

Including a runtime profiling analysis of KVTuner’s selection method would strengthen claims about efficiency.

**Other Strengths And Weaknesses:**

The paper makes an important contribution to memory-efficient LLM inference with strong empirical results. However, there are areas for improvement:

1) The additional computational cost of  computation is not fully analyzed.

2) The practical impact on multi-head attention efficiency is unclear.

3) The method’s effectiveness in extremely long-context settings (e.g., 100K+ tokens) is not evaluated.

**Questions For Authors:**

What is the additional FLOP overhead per generation step compared to traditional KV quantization methods?

How does KVTuner scale with batch size increases?

Can KVTuner be integrated with KV cache eviction methods like SnapKV for improved memory efficiency? Have you considered hybrid approaches?

How does KVTuner handle extreme long-context inference (e.g., 100K+ tokens)? Does performance degrade due to accumulated quantization errors?

**Relation To Broader Scientific Literature:**

The paper is well-situated in the literature on KV cache quantization and memory-efficient LLM inference. It correctly cites works on uniform KV quantization (KV8, KV4) and hybrid eviction strategies. However, as discussed in lines 275-295, it does not sufficiently compare with recent approaches that integrate quantization with eviction (e.g., SnapKV. A direct comparison would strengthen the positioning of KVTuner as a practical alternative to existing methods.

**Theoretical Claims:**

The paper correctly identifies that key cache quantization errors accumulate across both model layers and generation steps, leading to significant degradation in long-context inference. The discussion in lines 330-350 formalizes the optimization problem for selecting layer-wise precision pairs, but it does not analyze whether the proposed selection strategy guarantees global optimality. Additionally, while KVTuner reduces memory usage, it does not completely eliminate online computational overhead.

---

> ### Author Rebuttal · Authors · 2025-04-01
>
> We sincerely thank you for the thoughtful feedback and constructive critiques. Below, we address each concern and outline planned revisions to strengthen the paper:
>
> ---
> # 1. Computational cost
> **The profiling and layer-wise KV cache precision tuning are completely offline and no online overhead for precision selection is introduced.** Only online quantization for the newly generated key and value token using the calibrated precision similar to uniform KV precision is applied.
>
> We only perform 200 rounds of offline multi-objective optimization search with limited data, which is efficient and **one-time cost**. This is a huge advantage compared with other mixed-precision approaches that incorporate token-importance ranking for KV selection.
> **The two-stage search space pruning only takes seconds.** The MOO search with pruned search spaces is the most time consumping part, which performs 200 rounds of the target LLM on the selected 200 prompts. The optuna framework is also quite efficient. **The simple huggingface transformers may takes hours** in the above settings with single batch inference in Nvidia RTX 4090. The offline tuning time cost is acceptable compared with the saved cost during large-scale online service and model pretraining cost. In addition, better hardwares, torch graph compiling, and multi-batch inference during tuning may reduce the tuning cost to less than one hour.
>
> ---
> # 2. Latency experiment
> The layer-wise FLOP cost difference is mainly caused by efficiency difference of the KV cache precision pairs.  The model-level efficiency reflects the overall effects of layer-wise efficiency of all KV cache precision pairs. The memory movement cost from CPU sto GPUs linearly increases with the KV cache size in most case and attention is normally memory bounded. We also report the total model-level throughput comparison of Llama-3.1-8B-Instruct using the searched config in Table 5 as below.  The hardwares are Nvidia RTX 4090 24G. **Compared with KIVI-KV8, the throuhgput of KVTuner-C3.25 can be improved by 16.79%~21.25%.**
> |BS, inputLen|KV8(baseline)|K8V4|KV4|K4V2|KVTuner-C4.92|KVTuner-C3.25|
> |-|-|-|-|-|-|-|
> |64,128|3836|4193|4567|4697|4240(10.53%)|4652(**21.25%**)|
> |16,512|1102|1205|1275|1304|1239(12.41%)|1296(**17.55%**)|
> |8,1024|549|597|632|645|600(9.22%)|641(**16.79%**)|
>
> ---
> # 3. Long context effectivenss
> We compare KVTuner with the basedlines KIVI-8, KIVI-4, our proposed variant KIVI-K8V4, and per-token-asym ones in the 20 Longbench datasets and the averaged scores are as below.  **KVTuner pushes KV cache quantization for the nearly lossless long context generation to 3.25-bit, outperforming the uniform KV precision.**
> ### Qwen2.5-7B-Instruct KIVI
>  |BF16|KIVI-8|KIVI-K8V4|KIVI-4|KVTuner-C4.92|KVTuner-C3.25|
> |-|-|-|-|-|-|
> |0.7956|0.7992|0.8001|0.7723|0.7956|0.7903|
>
> ### Qwen2.5-7B-Instruct per-token-asym
>
> |BF16|KV8|K8V4|KV4|KVTuner-C5.0|KVTuner-C4.0|
> |-|-|-|-|-|-|
> |0.7956|0.7971|0.7953|0.6343|0.8005|0.7960|
>
> ---
> # 3. Global optimum
> Due to the complex and nonlinear dependency of error accumulation, there are no theoretically global optimization solutions for the NP-hard problem. The MOO formulation balances Pareto-optimal solutions rather than seeking global optimality. The two-stage search space pruning aims to help MOO converge and reduce searching cost, comparing Figure 5 and 6.
>
> ---
> # 4. Scaling over batch size
> When the input and output sequence lengths are fixed at 512 and 128, respectively, the following table presents the throughput (in Token/s) of KVTuner.
>
> |BS|4|8|16|32|64|80|
> |-|-|-|-|-|-|-|
> |Tokens/s|494|938|1776|2889|4655|4964|
>
> ---
> # 5. Integration with eviction
> We agree that integrating with eviction methods is promising and leave it in our future work. KVTuner is fully compatible with KV cache eviction methods including StreamingLLM, H2O, and SnapKV, because quantization and eviction are two orthogonal approaches.
>
> ---
> # 6. MHA
> We mainly test on LLMs with grouped query attention (GQA), because most recent and powerful models are GQA based and GQA is a variant of MHA. The layer-wise sensitivity to KV cache quantization is the inherent property of LLM with multi-layer transformers. In addition, key and value in attention layers have different sensitivity to quantization. There are the only two assumptions of KVTuner. We also analyze MLA models such as Deepseek-v2-lite-chat, in which we also observed similar layer-wise patterns.
>
> ---
> # Conclusion
> KVTuner’s practical impact lies in its deployability: it requires no inference-time overhead and achieves near-lossless accuracy, making it a compelling solution for production LLM systems and most LLM acceleration hardwares.
> In addition, we study the underlying mechasim of the higher importance of key cache. We also theoretically analyze that the layer-wise sensivity of attention heads to KV cache quantization strongly correlates with attention patterns, which is novel and may provide more insights to model design and compression.

---

### Decision · Program_Chairs · 2025-05-01

**Decision:**

Accept (poster)

**Comment:**

This paper studies KV cache efficiency, a critical topic for today's LLM research. It provides an adaptive framework that prioritizes the key cache over the value cache for quantization, which brings additional performance improvements.

While the reviews are mixed, the rebuttal somewhat alleviated some of the concerns with one reviewer raising their rating. I agree with most of the reviewers that this result can benefit the community due to the importance of the topic and the promising experimental results. Thus, I recommend weak acceptance. If the paper is accepted, I strongly advise the authors to address all concerns raised by the reviewers.